# Results of Field Experiments for the Creation of Artificial Updrafts and Clouds

Magomet T. Abshaev [1], Ali M. Abshaev [1,*], Andrey A. Aksenov [2,3], Julia V. Fisher [3], Alexander E. Shchelyaev [3], Abdulla Al Mandous [4], Omar Al Yazeedi [4], Youssef Wehbe [4], Emil Sîrbu [5], Dragoș Andrei Sîrbu [5] and Serghei Eremeico [5]

1 Hail Suppression Research Center "Antigrad", 360004 Nalchik, Russia
2 Joint Institute for High Temperatures, Russian Academy of Sciences, 125412 Moscow, Russia
3 Engineering Company TESIS, 127083 Moscow, Russia
4 National Center of Meteorology, Abu Dhabi P.O. Box 4815, United Arab Emirates
5 S.C. Intervenții Active în Atmosferă S.A., 031827 Bucharest, Romania
* Correspondence: abshaev.ali@hsrc-antigrad.com

**Abstract:** This study documents results from a series of field experiments on the creation of artificial updrafts and convective clouds at a test site in the United Arab Emirates (UAE). The proposed method incorporates a vertically directed jet from an aircraft turbojet engine saturated with active hygroscopic aerosols for the purpose of energetically feeding the jet with water vapor condensation heat below cloud base level. This paper presents the description and main characteristics of the experimental equipment, methodology of experiments and atmospheric conditions, analysis of the obtained results, and prospects for further development of the proposed method. On the whole, the experiments showed that under the conditions of low air humidity, typical for the UAE, and the slowness of the condensation process, the replenishment of the jet energy by the heat of condensation is too small, and the power of the used jet engine in the experiments is insufficient to overcome surface temperature inversions, horizontal winds, and initiation of deep convection. Nevertheless, the results of field experiments and numerical simulation made it possible to outline promising directions for further research on improving the considered method for creating artificial clouds and precipitation.

**Keywords:** turbojet engine; artificial updrafts and clouds; hygroscopic aerosol; condensation heat; atmospheric conditions; field experiments; results and prospects

## 1. Introduction

In many regions, the shortage of fresh water stimulates the search for non-traditional water resources [1–3] and the development of artificial precipitation enhancement technologies based on the seeding of clouds with glaciogenic and hygroscopic agents [4–6]. The effectiveness of these technologies depends on many factors, including cloud resources. In arid and desert regions, cloud resources are extremely limited [7,8]. Therefore, the purpose of this research, carried out within the framework of the UAE Research Program for Rain Enhancement Science (UAEREP) [9], is to study the possibility of creating artificial clouds.

It is known that specific cloud types form under certain atmospheric conditions and anthropogenic forcing. Pyroclouds form over forest and other large fires, active volcanoes, and sun-heated mountain slopes [10–12], while "urban clouds" or sometimes-called "industry clouds" form over megacities and large energy objects (nuclear and thermal power plants, oil refineries) with large thermal energy emissions [13]. These clouds are caused by thermal convection initiated by these heat sources. They can sometimes reach the thunderstorm stage, but are usually not predicted by standard weather forecasts.

Large fires can lead to the formation of powerful cumulonimbus clouds with significant precipitation. Numerical simulation of thermal convection over forest fires in the suburbs

of Canberra (Australia), where a series of large pyro-cumulonimbus clouds was formed, showed that in addition to the heat flow, water vapor plays a major role in the development of convection. Without a moisture source, the "dry" convective plume reaches a height of about 7 km, while in the simulation of heat and moisture sources, the cloud reaches a height of 14 km and can generate rainfall, hail, and even tornadoes.

Formation of pyroclouds above craters of volcanoes is caused by the rise of air heated over boiling lava even in the absence of an eruption. For example, the active volcano Nyiragongo in the Congo, which has erupted 34 times since 1882, has a two-kilometer crater filled with a lake of boiling lava, which initiates thermal convection almost constantly. In the case of volcanic eruptions, thousands of cubic kilometers of high temperature gasses and hundreds of kilometers of hot lava can be emitted into the atmosphere, which leads to the formation of large-scale cumulonimbus clouds, the peaks of which often penetrate the stratosphere.

Convective clouds over the mountains are common on clear sunny days. Mountains tend to heat faster from the sun compared to lower-altitude terrain, and convective clouds are formed as they warm up, even on days with stable stratification. This is facilitated by the proximity of the mountain peaks to the level of condensation, as well as mountain–valley circulation and in the case of coastal mountains, the breeze circulation. Often, such clouds generate rainfall, growing to the scale of thunderstorms and even hailstorms, although their prediction is still challenging.

There are also cases of the development of pyroclouds over large fires associated with military operations. For example, on the night of 27 July 1943, after the dropping of 1300 tons of high explosives and 1100 tons of incendiary bombs on the city of Hamburg [14], a huge fire blazed up, over which a powerful cumulonimbus cloud with rain appeared and extinguished the fire [11]. According to the description of H. Dessens [15], rain developed in a situation when weather conditions normally prevent the development of rain. A powerful rainy cloud also formed after the nuclear explosion in Nagasaki.

Most often, urban clouds are observed over megacities. Chandler T.J. et al. [16] showed that the higher the density and area of multilevel buildings, the more intensive is the island of heat. In Moscow [17], New York, and Tokyo, the near-ground air temperature difference between the center and the suburbs reaches 10 degrees. On the vertical axis of the island, heat can spread up to 1.5 km, leading to the development of thermal convection and formation of clouds, increasing the frequency, intensity, duration, and amount of precipitation [13,18–20] and other storm properties [21]. Studying the climatology of the city near six large cities in the USA, Changnon [22] found an increase in the number of hail days from 2 to 4 times, an increase in the size of hailstones to 2.5 times, an increase in the width and length of the hail paths by several times, and an increase in the hail kinetic energy. It also showed a significant increase in the amount of summer precipitation in the St. Louis region, falling from the lee of the city, where the urban heat trail is affected.

NASA researchers Marshall et al. [23] used data from the Tropical Rainfall Measuring Mission (TRMM) satellite to confirm that urban heat islands create more summer rains over the major cities of Atlanta, Dallas, San Antonio, and Nashville on their leeward side. They found that the average monthly rainfall within 30–60 km by the wind was on average 28% more than on the windward side of the metropolis [24]. The maximum rainfall in the leeward areas often exceeded the maximum values in the windward regions by 48 to 116%. In some cities, the total rainfall increased by 51% [25]. These results are in agreement with earlier studies by Changnon [21] in St. Louis, Missouri, and not far from Atlanta.

Cities with a dimension of 30 to 50 km are powerful sources of heat, capable of creating new urban clouds and strengthening existing clouds. According to satellite images, Gretchen Cook-Anderson [26] found a significant effect of urban thermal island on plant vegetation at a distance of up to 10 km from the city boundaries and lengthening the growing seasons in 70 cities of eastern North America for 15 days compared to rural areas outside the city's influence.

Ground radar and rainwater networks detect an increase in the frequency and amount of precipitation over large nuclear power plants, thermal power plants, and refineries [27]. The heat and hot water vapor emitted by these objects create a local low-pressure zone that leads to a convergence of moist air, updrafts, and formation of clouds, increasing the frequency and intensity of showers and thunderstorms [28].

These facts testify to the fundamental possibility of creating artificial clouds by heating surface air and initiating thermal convection which reaches the level of condensation. There are a number of attempts to implement this.

Analysis of the results of previous theoretical and experimental research on creation of artificial convective clouds showed that they are all based on the stimulation of thermal convection using different heat sources (man-made fires, flare, and jet meteotrons) which heat local areas of the surface atmosphere:

i. The ancient method of causing rain during drought, involving the creation of artificial fires in the prairies and savannas, used in South America and Equatorial Africa.

ii. Cloud creation using meteotrons that are artificial heat sources warming the surface air by the heat released during the combustion of petroleum products.

    ii.1. Meteotrons used in France and in Cuba [14,15] to create clouds, containing 100 or more jet burners, placed on an area with a radius of 33 m. About 60 and 105 tons of gas oil were burned per hour and generated energy capacity reached 700 and 1000 MW, respectively.

    ii.2. The meteotron of the Institute of Geology and Geophysics of the Siberian branch of the Russian Academy of Sciences had 60 jet flamethrowers placed around the perimeter of the octagon with a side of 53 m and developed a capacity of 5000–6000 MW, spending about 430 tons/h of diesel fuel. Updrafts and black smoke in some of the eight experiments rose to 3 km [29].

    ii.3. Meteotron of the Institute of Applied Geophysics of USSR Hydrometeorological Service with four and ten jet engines, designed to study the possibility of creating artificial clouds and precipitation, had power of 200 and 500 MW, and "supermeteotron", built on the shore of the alpine lake Sevan in Armenia in the hope of replenishing the water level in it, contained six engines with a total power of 500 MW [29].

    ii.4. Meteotrons of Chelyabinsk Polytechnic Institute (eight variants) were designed for the ventilation of coal mines, creating clouds and fog dispersion. They contained from 10 to 100 centrifugal injectors with a diesel fuel consumption of seven to 30 tons/h, and developed a capacity of 80 to 400 MW. Fine spray of diesel fuel in combination with various nozzles provided complete combustion of fuel and resulted in a smokeless jet [30,31]. The experiments showed that a meteotron with power of about 1000 MW can create an updraft reaching a height of 100–1000 m or enhance existing clouds, and under favorable conditions, noted the development of a small convective cloud.

    ii.5. NASA Mississippi Space Center tested the super heavy rocket engine RS-68 and demonstrated the generation of an artificial cloud that produced light rain (from open Internet sources). This was facilitated by the intense release of hot water vapor, which cools by raising and condenses as the cloud.

iii. Proposals to stimulate the development of clouds with solar meteotrons are also known:

    iii.1. The simplest solar meteotron is the area of earth's surface covered with asphalt, black cloth, or other materials that absorb solar radiation.

    iii.2. Oranovsky solar meteotron [32] consisted of a blackened screen raised over the ground and surrounding mirrors, focusing on the additional solar energy generated by the screen.

    iii.3. Methods that reduce the albedo of the underlying surface by planting greenery and other bio-geoengineering methods. Branch and Wulfmeyer [33] believe that a viable ecological perspective to solve the problem of increasing

precipitation in arid and semi-arid regions is agroforestry. Brenig et al. [34] believe that the air over a surface with a low albedo, more heated by the sun's rays, rises and leads to the rise of the sea breeze. This contributes to the development of clouds and increased rain on the leeward side through 25–30 km inland.

iii.4. Method of creating an artificial aerosol layer in the near-ground atmosphere that absorbs solar radiation and can lead to the heating of local air volumes and initiate thermal convection by using highly efficient smoke compositions [35]. The disadvantages of these methods are the following:

- The method of burning the vegetation of the prairie and savanna is not acceptable due to their limited availability and adverse effects on flora and fauna.
- Torchlight and fire meteotron requires a large expenditure of fuel, which generates drastic environmental pollution.
- Coating of asphalt and black materials with a low albedo of large areas is not cost-effective, and their efficiency reduces energy loss due to leakage into the soil.
- Method of Oranovsky reduces heat losses due to rising of the blackened screen from soil on special supports. However, its practical realization is overly complicated due to the excessively large dimensions of blackened screen and mirrors involved (several $km^2$).
- Solar meteotron absorption screens in the form of asphalt, fabric, and other black coatings with small areas is not effective, and the creation of screens having an area of about 2–6 $km^2$ or more is very expensive and still at the conceptual level.
- Creation of artificial aerosol layer over land area 1–10 $km^2$ for at least 30 min requires consumption of several tens of tons of pyrotechnic composition and is applicable only away from populated areas and under specific weather conditions.
- None of these methods of creating ascending streams has successfully initiated the development of artificial clouds of practical importance.
- This work attempts to answer whether it is possible to overcome the limitations of these methods by energetically feeding the upward flow created by a vertically directed jet of an aircraft engine, due to the heat of condensation of water vapor on a hygroscopic aerosol introduced into the jet.

The purpose of this work is to experimentally verify the possibility of creating artificial clouds in an arid region based on a new method involving introduction into the atmosphere of a vertically directed jet fed by the heat of water vapor condensation on a hygroscopic aerosol injected into the jet at the start.

To this end, Section 1 of this paper gives an overview of the capabilities of previously used methods of creating artificial clouds which did not give practically acceptable results. Section 2 gives the physical principle and a brief description of the proposed method of creating artificial clouds which was tested in field experiments.

Section 3 contains a description of the experimental complex, including a description of the device for creating artificial updrafts on the basis of an aircraft jet engine, a description of devices for introducing hygroscopic substances into the jet, and equipment for instrumental control of the experiments.

Section 4 is devoted to the organization of field experiments for testing the proposed method and device, including algorithms for selecting days and times of day with favorable atmospheric conditions, the methodology for conducting experiments, and a description of meteorological conditions for conducting experiments during two field campaigns in the UAE. Section 5 describes the results of the experiments, including instrumental data on the artificial updrafts and a description of the physical effects observed during the experiments.

Section 6 discusses the results of the experiments, which also did not lead to the creation of artificial clouds in the arid conditions of the UAE, and the reasons for this are considered. Section 7 gives conclusions and recommendations on improvement of the tested method and device.

## 2. Method for Creating Artificial Updrafts and Clouds

Based on the analysis of the performance of various physical principles for stimulating convection in a cloudless atmosphere, a new method for creating updrafts and artificial clouds was proposed [36–39], which is based on stimulating an updraft using a vertically directed jet and jet energy supply due to the heat of water vapor condensation on three types of hygroscopic aerosols introduced into the jet at the start. It is assumed that, in contrast to the Vulfson and Levin [29] meteotrons, in which the buoyancy of the jet rapidly decreases as it rises in the atmosphere, in the proposed method, feeding the jet with the heat of water vapor condensation can increase its buoyancy and provide rise to the level of cloud formation. To stimulate the condensation of water vapor at the start of the jet, three types of hygroscopic aerosol are introduced, which have different hygroscopic points ($h_{gp1} < 6\%$, $41 < h_{gp2} < 70\%$, and $71 < h_{gp3} < 80\%$), which can provide the jet with condensation heat at any air humidity.

The hygroscopic point ($h_{gp}$) of hygroscopic solid material can be identified through its water uptake capacity and represents the threshold value of the relative humidity in the air above which the solid substance starts adsorbing water vapor [40].

The choice of hygroscopic substances was carried out taking into account their hygroscopic and thermodynamic properties, and environmental and fire safety. Given that the processes of dissolution (hydration) and crystallization are accompanied by thermal effects, it is desirable that the selected substances release heat during deliquescence and absorb heat during crystallization. Taking into account these factors, calcium chloride $CaCl_2$, urea $(NH_2)_2CO$, and edible salt, $NaCl$, were chosen as working substances.

One of the options for creating aerosols of each type was the method of spraying aqueous solutions of the selected substances using nozzle systems to droplets about 15 μm in diameter, the evaporation of which forms an aerosol with a diameter of about 5–10 μm. The second method of preparing aerosols involved fine grinding of granules.

During field experiments, along with solution spraying, powders of the listed substances were used, as well as $NaCl/TiO_2$ novel hygroscopic micro-powder, which is capable of condensing much more water vapor than the listed types of aerosols [41–43].

When air humidity exceeds the hygroscopic point, a saturated solution droplet first forms on the aerosol particle, above which the water vapor pressure $E_S$ is lower than in the environment $E_\infty$. This leads to further droplet growth due to water vapor diffusion until $\Delta E_S = (E_\infty - E_S) > 0$. We found [36,38] that the mass of water vapor $m_w$ that can condense on an aerosol of mass $m_a$ can be calculated by the Equation:

$$\frac{m_w}{m_a} = \left[ (1 + \kappa)\frac{C_c}{C_s} - 1 \right] \tag{1}$$

where $m_a$ is the mass of a dry aerosol particle; $k$—ratio of the mass of water to the mass of the dissolved substance in a saturated solution; $C_C = E_S/E$—ratio of water vapor pressure over a drop of saturated solution to the pressure over a flat surface of distilled water; $C_S = (\Delta Es)/E_\infty$—ratio of the difference between the indicated pressures to the pressure above the flat surface of distilled water.

The values of $\kappa$, $C_C$, and $C_S$ can be found in reference books on saturated solutions [40]. For example, at 20 °C in a saturated solution of $NaCl$, there are 37.1 g of $NaCl$ per 100 g of water. The value $\kappa = 100/37.1 = 2.8$, value $C_C = 0.78$, and $C_S = 0.22$. According to Equation (1), an aerosol particle of $NaCl$ can condense $k_1 = m_w/m_a = 12.47$ times its dry mass.

The hygroscopic growth factor is often expressed by the ratio $D_d/D_a$, where $D_a$ is the diameter of the dry aerosol and $D_d$ is the diameter of the formed droplet. Values of $D_d/D_a$ can be calculated by Köhler's theory [43]. Experimental studies of $D_d/D_a$ dependence

on air humidity on the Low-Temperature Hygroscopicity Tandem Differential Mobility Analyzer [44,45] have shown that at $RH \geq 90\%$ and 20 °C, the NaCl particles of diameter $D_a$ = 0.1 μm produce droplets $D_d$ = 2.35·$D_a$. New experiments by Bermeo et al. [46] showed that the growth factor of NaCl particles with a size of about 0.8 μm is $D_d/D_a$ = 2.26.

Differences in the values of growth factor $D_d/D_a$, as well as the ratio of volumes $V_d/V_a$ and masses $m_d/m_a$, of formed droplets obtained by different authors have a noticeable scatter (see Table 1). This scatter is apparently due to the inaccuracy of the experimental values of $D_a$ associated with the non-sphericity of dry aerosol, the presence of cracks and cavities in its structure, and small but different water content. Further, this paper uses the results of measurements by Bermeo et al. [47].

**Table 1.** NaCl aerosol growth factor at RH $\geq$ 90%.

| Author | $D_a$ (μm) | Ratio $D_d/D_a$ | Ratio $V_d/V_a$ | Ratio $k_1 = m_d/m_a$ | Droplet Density, $\rho_d$ (g/cm³) |
|---|---|---|---|---|---|
| Köller's theory [43] | 1 | 2.98 | 26.46 | 12.76 | 1.085 |
| Equation (1) Abshaev et al. [36] | 1 | 2.97 | 26.2 | 12.47 | 1.09 |
| Gysel's experiments [45] | 0.1 | 2.35 | 12.98 | 6.53 | 1.09 |
| Experiments by Bermeo et al. [46] | 0.8 | 2.26 | 11.54 | 5.86 | 1.10 |
| NaCl/TiO$_2$ aerosol growth factor at RH $\geq$ 90%. | | | | | |
| Experiments by Bermeo et al. [46] | 0.6 | 3.44 | 40.71 | 19.14 | 1.02 |

Estimations showed (see Tables 2–4) that at values of drop growth factors on hygroscopic particles indicated in Table 1, quite a significant amount of condensation heat is released. Feeding the jet with this amount of heat may have a tangible effect on its movement in the atmosphere.

A theoretical study of the possibility of creating artificial clouds using a vertically directed buoyancy jet saturated with hygroscopic aerosol was carried out [36] on the basis of adaptation and testing of the FlowVision computational fluid-dynamics software package suite [48,49]. A system of equations in a three-dimensional space was solved, including the Navier–Stokes equation, the energy equation, the vapor mass transfer equation, and the continuity equation [34]. Numerical experiments were carried out to study the structure of the velocity and temperature fields, as well as the concentration of the aerosol introduced into the jet, for different vertical profiles of wind speed, temperature, and air humidity.

The numerical simulation results showed that the motion of a high-speed jet in a real atmosphere has a complex turbulent character due to its high speed. The fields of the jet vertical velocity and excess temperature are extremely inhomogeneous even in a windless atmosphere [36].

As it rises, the jet expands, due to the entrainment of ambient air, and loses superheat and the rate of ascent. In this case, the jet temperature decreases faster than the speed. As a result, the upper part of the jet continues to rise by inertia even when its temperature equals the ambient air temperature, and then becomes even lower by 1 °C [37]. Spraying water and aqueous solutions of hygroscopic substances leads to a more rapid expansion of the jet and a decrease in its speed and temperature.

The wind in the surface atmosphere leads to jet deformation in the vertical and horizontal plane, tilting the updraft top to the leeward side for hundreds of meters (see Figure 1). The presence of even shallow wind ($U$ = 1 + 0.005 $h$), where $h$ is a height, almost halves the jet height, which greatly limits the possibility of creating artificial clouds. This is one of the reasons for the modest results previously obtained [29,50] in experiments with meteotrons.

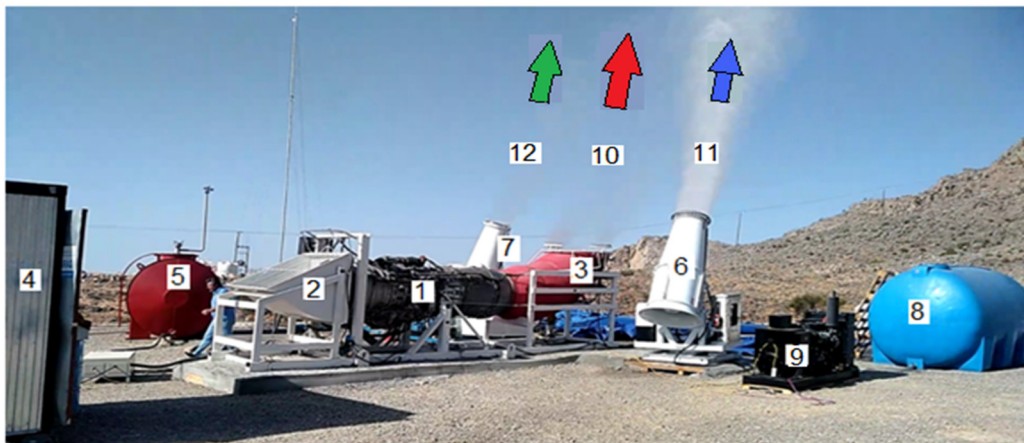

**Figure 1.** Experimental equipment: **1**—turbojet aircraft engine D-30; **2**—engine air intake unit; **3**—unit for the output jet vertical direction; **4**—engine start and control system; **5**—jet fuel tank; **6**—fog cannon JY-60; **7**—fog cannon WP-60; **8**—water solution tank for dissolved hygroscopic substances; **9**—diesel generator $3 \times 380$ V 50 kW, feeding fog cannon JY-60; **10**—high-temperature high-speed flow, generated by turbo jet engine; **11**—gas-droplet flow generated by fog cannon JY-60; **12**—gas-droplet flow generated by fog cannon WP-60.

The jet elevation increases as the temperature lapse rate increases from $\gamma = 6.5$ to 9.5 °C/km. For example, the jet rise height without recharge at $\gamma = 6.5$ °C/km does not exceed 420 m, and at $\gamma = 9.5$ °C/km, the jet rises to 840 m (Figure 1).

Inclusion feed of the jet by condensation heat can lead to a significant increase in the jet lift and its resistance to the destructive effect of the wind. The condensation heat released by the action of NaCl aerosol can cause the temperature of the jet to increase by about 1 degree and the velocity of upward flow by up to 1.5 m/s. Introduction of three types of aerosols into the jet can increase the jet lift by 14%. More effective may be the introduction of two-layer synthesized core/shell hygroscopic $NaCl/TiO_2$ aerosol [41–43], which can absorb about 19 times more water vapor of its dry mass (Table 1) and increase the amount of feeding energy by about four times compared to the same aerosol of pure NaCl. This increase in the feeding energy begins at humidity RH > 62% [43] and can increase the lifting power of the jet and the potential of creating an artificial cloud in the real atmosphere.

On the basis of numerical experiments, it was also established that an increase in the jet engine installation height above sea level, for example, by 1000 m, leads to an additional increase in the jet rise height, which is apparently explained by a decrease in the resistance of the medium and an increase in the temperature difference between the jet and the medium, where the higher the colder.

An increase in air humidity also leads to an increase in the jet lifting height even without condensation of water vapor, since moist air is lighter than dry air.

Therefore, numerical and experimental modeling of a vertically directed jet fed by the heat of condensation, as described in [39], is aimed to assess the probability of reaching the condensation level. This can, under certain atmospheric conditions, lead to the formation of convective clouds with precipitation.

## 3. Experimental Equipment

For carrying out field experiments on the creation of artificial updrafts and clouds, an experimental complex was prepared to release a vertically directed high-temperature jet, in which hygroscopic aerosols were sprayed to contribute to the condensation of water vapor and increase the released energy.

The experimental complex (Figure 1) contains:

- D-30 turbojet aircraft engine (1), air intake (2), elbow nozzle (3), launch/control system (4), fuel system (5);

- Aqueous solution-spraying system of the first type, which is a system of nozzles installed along the perimeter of the jet turning device;
- Two Fog Cannons (6 and 7) for spraying aerosols of the second and third types with fans having power of at least 20 kW, autonomous control, and power supply systems;
- System for the preparation and storage of aqueous solutions of hygroscopic substances, consisting of a mixer with a capacity of 1 m$^3$ and three tanks with a capacity of 3 and 6 m$^3$ (8);
- Mill for grinding granules of hygroscopic substances (not shown in the figure);
- Set of instruments for monitoring the results of field experiments.

The jet machine and fog cannons are installed so that the stream of heated gasses produced by the jet engine and the gas-and-drop streams produced by the fog cannons can merge into a unified gas-and-drop updraft.

The air intake of the turbojet engine (item 2 in Figure 1) has a rectangular section at the inlet, which then turns into a round one and is connected to the engine inlet using a round flange. The air intake area is two times larger than the engine inlet area to eliminate the lack of airflow. The air intake inlet is equipped with a mesh fence to ensure the safety of personnel and prevent birds and other bodies from being sucked into the engine. A system of nozzles is installed above the mesh, which serves to spray water and cool the air compressed in the compressor in order to increase the compression pressure and consequently increase the speed of the output flow. This system contains a Fogger-7 nozzle system with a nozzle diameter of 0.5 mm, a 10 Bar water pump, a water tank, hoses, and a faucet. A total of 80 nozzles provide atomization of about 600 g/s of water, increasing the moisture content of reactive gasses by 3 g/(m$^3$ s).

The *device for turning the jet to the zenith* (Figure 2) contains a cone-shaped diffuser and an outlet that serves to expand and turn the jet. To reduce jet energy losses, the outlet section of the diffuser is two times larger than the inlet. The outlet is made with a bending radius twice the radius of the pipe, the angle of rotation is limited to 70°, and the outlet of the pipe is cut horizontally to ensure the outflow of the jet upwards. This design makes the supersonic flow subsonic, weakening the strength requirements for the installation. Taking into account the losses in the turning device, the jet velocity at the exit to the atmosphere is about 300 m/s, its temperature rise above the environment is about 300 °C, and the mass flow rate of reactive gasses in the nominal engine operating mode is about 200 kg/s.

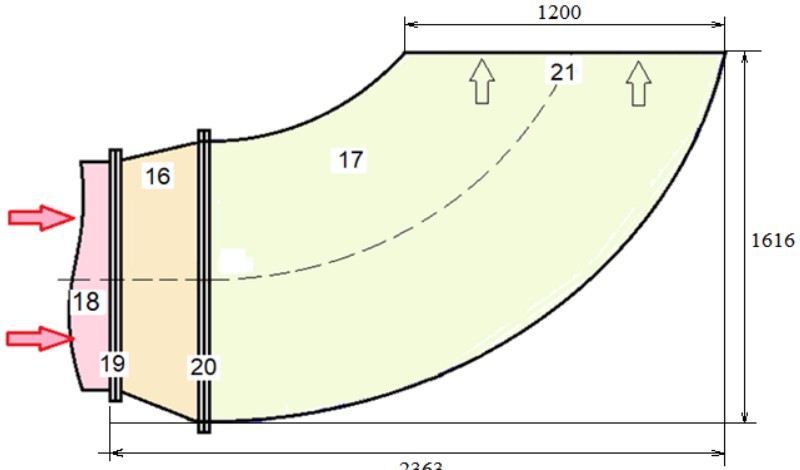

**Figure 2.** Device for turning the jet stream to the zenith: **16**—the cone-shaped diffuser; **17**—the 70° bend; **18**—engine nozzle outlet; **19** and **20**—connecting flanges; **21**—outlet of the bend, cut in horizontal plane.

The *system for spraying aqueous solutions of three hygroscopic substances* consists of a jet outlet atomizer and two fog cannons (items 6 and 7 in Figure 1).

The NaCl solution atomizer is mounted on an annular pipe surrounding the jet engine outlet and contains: 80 "Fogger-9" nozzles, an Espa Multi35 8N pump with power 4.9 kW, tanks for NaCl aqueous solution with volume 10 m$^3$, fine filter, pressure gauge, hose system, and valve.

*Fog cannons JY-60 and WP-60* with nozzle systems, powerful fans, and autonomous control and power systems, are used to spray $CaCl_2$ and $(NH_2)_2CO$ aqueous solutions and create high-speed gas-drop jets (items 8 and 9 in Figure 1) merging at a height of 20–30 m with main jet produced by aviation engine. These fog cannons are designed for dust suppression, plant spraying, and artificial snow making. They were modified to eject almost to the zenith (vertical angle $85 \pm 5°$) and merge with the main jet, and had 80–100 nozzles with diameter of 0.3–0.5 mm, which provide spraying of about 1 L/s of aqueous solutions into droplets with a size of 15 μm. The height of the gas-droplet jet ascend of this cannon depends on the spray pressure and fan power. For example, at a pressure of 3 MPa and a fan motor power of about 25–30 kW in a calm atmosphere, the jet reaches 60–80 m horizontally or about 40 m vertically.

Tests have shown that the JY-60 fog cannon sprays about 1.5 L/s of $CaCl_2$ aqueous solution, and the WP-60 cannon sprays about 0.8 L/s of $(NH_2)_2CO$ solution. Drops of the solution are ejected into the jet engine, where they instantly evaporate and form an aerosol. According to the CEM DT-9880 dust particle counter, the aerosol size varies from 0.3 μm to 10 μm.

*The system for the preparation and storage of aqueous solutions* contains:

- Mixer with a capacity of 1 m$^3$ for the preparation of highly concentrated aqueous solutions of these substances;
- Water tank with a capacity of 20 m$^3$;
- Three tanks for aqueous solutions of three different hygroscopic substances.

*Note*: The dissolution of $CaCl_2$ is accompanied by an abundant release of heat (674 kJ/kg), and the dissolution of $(NH_2)_2CO$ by the absorption of 274 kJ/kg. Therefore, solutions of the required concentration were prepared in stages, taking into account the time of cooling or heating the solutions.

The fundamental difference of this experimental system from previously used meteotrons is that a high concentration of three types of coarsely dispersed hygroscopic aerosols is introduced into the jet in order to initiate the condensation of water vapor and increase the jet buoyancy by the heat of water vapor condensation.

To control the experiments and their results, an instrumental complex was used that provides measurements of the main parameters of the atmosphere and an artificial jet (Table 2 and Figure 3).

**Table 2.** Equipment for instrumental control of experiments.

| Name | Measured Parameters |
|---|---|
| Weather station WXT-536, Vaisala, Vantaa, Finland | Temperature, pressure, humidity, precipitation, and wind at the ground |
| Multi-microwave radiometer RPG HATRO G4 PRO, RPG Radiometer Physics GmbH, Meckenheim, Germany | Vertical profiles of humidity, air temperature, vertically integrated liquid water, and vapor |
| Doppler pulse wind Lidar Halo Photonics Streamline XR | Doppler wind components and profiles of horizontal wind speed and wind direction (each 5 min) |
| Thermograph IRTIS-2000 C, IRTIS Ltd., Moscow, Russia | Measurement and visualization of temperature field deformations created by turbojet engine flow. Spectral range 3–5 μm, accuracy 1 °C, temperature range −50 to +500 °C. |
| Pyranometer RK200-03, Rika-Sensor, Changsha, China | Solar radiation flux |
| Digital thermometer TM902C | Jet temperature |
| GEM DT-9880, Shenzhen Everbest Machinery Industry Co. Ltd., Shenzhen, China | Aerosol concentration in six ranges: 0.3; 0.5; 1; 2.5; 5; 10 μm |
| Noise meter MEGEON 92131 | Jet noise level |
| Hexacopter weather drones, HSRC Antigrad, Nalchik, Russia | Air temperature, humidity, and pressure; 3D wind speed and direction, aerosol concentration |

**Table 2.** *Cont.*

| Name | Measured Parameters |
|---|---|
| Laser distance meter Extend LRS | Height and rate of rise of the cloud top |
| C-band weather radars | Radar parameters of artificial clouds and rain |
| Meteorological satellites Meteosat-10 | Visible and infrared imageries of clouds |
| Abu-Dhabi airport air sounding data http://weather.uwyo.edu (accessed on 21 January 2022) | Thermodynamic parameters of the atmosphere at altitudes from ground level to 35–40 km |

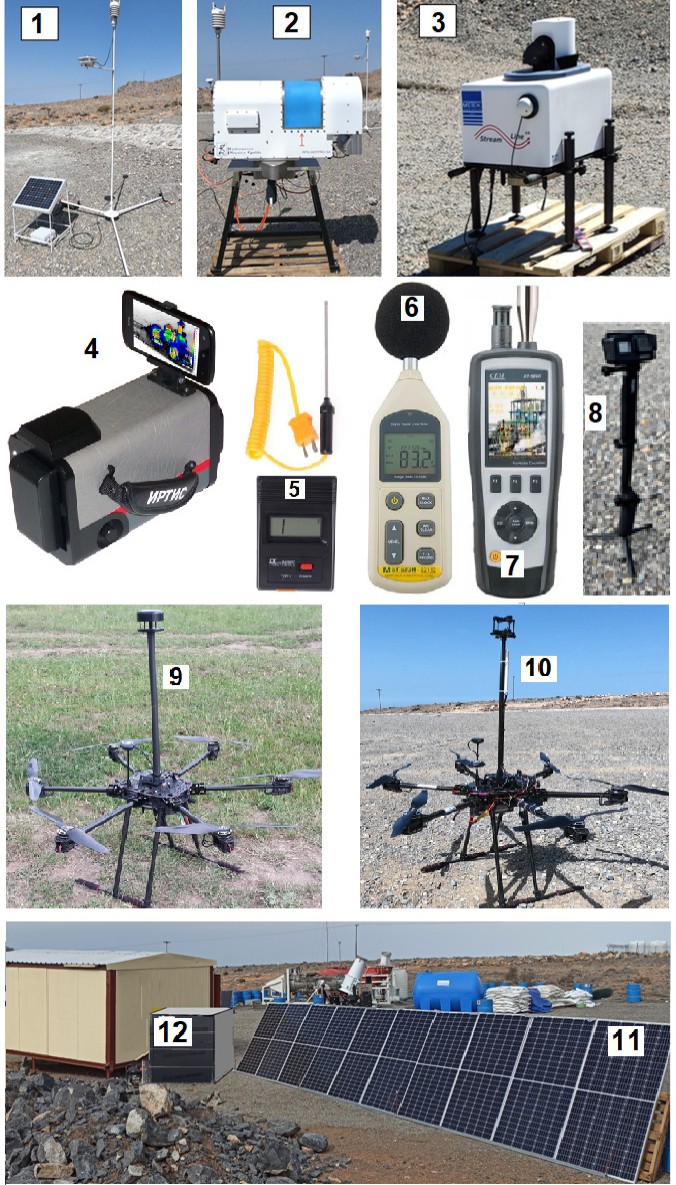

**Figure 3.** Instruments for measurement of atmospheric parameters and experiments control: **1**—automatic weather station Vaisala WXT-536 with pyranometerRK200-03; **2**—multi-microwave radiometer RPG HATPRO-G4 PRO; **3**—Wind Lidar Halo Photonics Streamline XR: **4**—Thermograph IRTIS-2000C for making infrared images of temperature disturbances; **5**—TM902C digital thermometer for measuring jet temperature; **6**—noise meter MEGEON 92131; **7**—aerosol meter CEM DT-9880; **8**—video camera; **9** and **10**—weather drones equipped with integrated temperature–pressure–humidity sensor, acoustic wind speed and direction sensor, and aerosol sensor; **11**—solar battery for round-the-clock power supply of the measuring complex; **12**—voltage converter.

## 4. Organization of Field Experiments

The purpose of the experiments was to test the possibility of creating artificial updrafts and convective clouds using a vertically directed jet saturated with three types of hygroscopic aerosol, introduced to stimulate the condensation of water vapor and increase the jet buoyancy due to the latent heat of condensation.

The experimental complex was mounted on Jebel Jais Mount on a site located at an altitude of 1600 m (25°56′40.55″ N 56°09′21.12″ E) on a saddle between two peaks, the height of which is about 250 m higher and located at a distance of 660 and 390 m from the peaks positions.

In the first field campaign from 17 February to 28 March 2021, four trial experiments were carried out, and in the second field campaign from 25 December 2021 to 15 January 2022, 15 experiments were carried out.

### 4.1. Selection of Days and Times of Day with Favorable Atmospheric Conditions

According to theoretical modeling data [36,38], initiation of the development of artificial convective clouds in a cloudless atmosphere is possible only on days with some favorable conditions, when:

- Surface wind speed $U \leq 2$ m/s and wind shear $dU/dh \leq 0.005\ h$;
- Temperature lapse rate $\gamma \geq 7.5$ °C/km;
- Air humidity $f > 66\%$;
- Absence of powerful temperature inversion layers;
- CAPE > 200 J/kg.

Initially, it was assumed that the decision to conduct experiments would be made on a randomized basis, but given that in the arid conditions of the UAE, the above favorable conditions can be quite rare, it was decided to conduct experiments on all days with more or less favorable conditions.

The choice of days with such conditions was carried out on the basis of measurements in real time of:

(a) Vertical profiles of wind speed and direction by Doppler Lidar Streamline XR and weather hexacopter drones with kit of sensors;
(b) Vertical profiles of air temperature and humidity over the experimental position by microwave radiometer RPG HATRO PRO;
(c) Wind speed, air temperature, and humidity at the site level by weather station Vaisala WXT-536.

Upper air sounding data from Abu Dhabi Airport, weather forecast, and Weather Radars Network of the UAE National Center of Meteorology (NCM) and geostationary satellite Meteosat-10 data on cloud fields in the region were also used.

The algorithm scheme for making a decision to conduct an experiment is shown in Figure 4.

In order to increase the likelihood of success, experiments to create artificial clouds were carried out during the hours of maximum natural convective activity of the atmosphere. It is known that the maximum solar radiation is observed at 12:00 local time, maximum temperature of the soil occurs in 1.5–2 h, maximum surface air temperature is usually observed at about 15:00 local time, and maximum amount of rainfall, thunderstorms, and hail observed, for example, in Russia is around 17:00 [51]. With this in mind, the experiments began in the period from 15:00 to 16:30 local time, except for cases when there were attempts to strengthen the clouds moving over the field site.

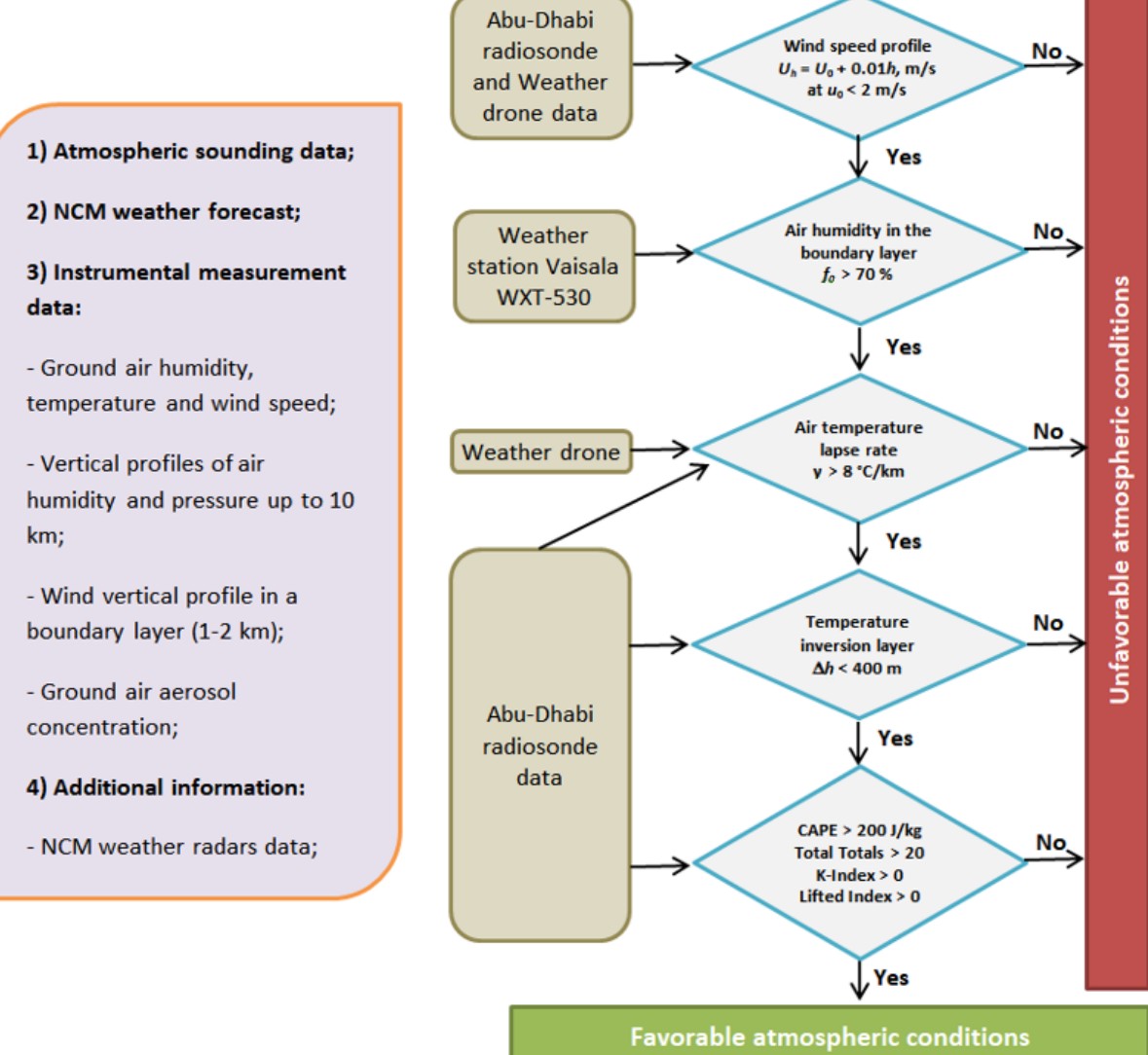

**Figure 4.** Criteria for choosing days with favorable conditions.

*4.2. Experiments Methodology*

Field experiments included the creation of an upward flow using a vertically directed jet of a turbojet engine saturated with coarsely dispersed hygroscopic aerosols having various hygroscopic points. It was assumed [36–39] that the introduction of these aerosols would lead to the condensation of water vapor and increase the energy of the jet due to the heat of condensation.

An aerosol of hygroscopic substances was formed by spraying aqueous solutions of $CaCl_2$, NaCl, and $(NH_2)_2CO$ into droplets 10–15 μm in diameter, upon evaporation of which aerosols 5–10 μm in size were formed. The solution flow rate was chosen so that at least $10^{11}$ particles/s of each type were introduced into the jet.

In some experiments, in addition to or instead of the solutions, $CaCl_2$ and $NaCl/TiO_2$ powders were introduced into the jet. $NaCl/TiO_2$ micro-powder was sourced through the UAE NCM. Due to its high hygroscopicity, $CaCl_2$ powder was prepared before experiments by grinding granules using an Akita JP 6SM-140D mill.

Before and during the experiments, the near-surface parameters (air temperature and humidity, wind speed, and direction), vertical gradients of air temperature and humidity, wind speed, and the parameters of the artificial updraft jet created by the turbojet engine (temperature, shape and height, the speed of the upward flow, the concentration of atmospheric and introduced aerosol, etc.) were recorded. Background aerosols 2 m above

surface were measured on a daily basis by hand aerosol meter CEM DT-9880 in six size bins—0.3; 0.5; 1.0; 2.5; 5.0 and 10 μm.

The jet parameters were controlled using professional infrared thermograph IRTIS-2000C and meteorological drones. The Thermograph allowed remote detecting of temperature and aerosol disturbances produced by the jet engine machine, while drones provided in situ measurements of temperature, humidity, pressure, wind speed (horizontal and vertical) and direction, and aerosol concentration along the flight trajectories.

The drones rose to a height of 300–500 m and flew over the jet stream in a zigzag or circular trajectory with a radius of 200–300 m to detect an artificial updraft. The updraft was considered fixed if the drone, in addition to vertical speed, detected bursts of temperature and aerosol concentration. The development of clouds was controlled with the help of photo and video observations, satellite, and radar observations. Table 2 lists the characteristics of each experiment date, while Table 3 lists the consumption of hygroscopic substances.

**Table 3.** Date, time, atmospheric conditions, and results of field experiments on the creation of artificial updrafts and clouds.

| No | Date, Time, Duration of Experiment (min) | Weather Conditions | Surface Atmosphere Parameters | Atmospheric Instability Indices | Seeding Materials Injected into Jet | Results |
|---|---|---|---|---|---|---|
| 1 | 15 March 2021 $15^2$–$15^{16}$, 14 | Cirrus-Stratus clouds 25% to the East | $T_0 = 26.1\,°C$, $f_0 = 14.5\%$, $U_0 = 1$–$3\,m/s$ | LI = 6.12; SWEAT = 33; KI = $-28.5.5$; VT = 27.9; TTI = 16.8; $CAPE_{uvt} = 0$; $H_{cond} = 1700\,m$ | Solutions of $CaCl_2$ 1.5 L/s; NaCl 0.6 L/s; Carbamide 0.8 L/s. | Translucent plume in the sky |
| 2 | 17 March 2021 $15^{15}$–$15^{30}$, 15 | Clear sky | $T_0 = 23.7\,°C$, $f_0 = 17.00\%$, $U_0 = 3$–$4\,m/s$ | LI = 4.13; SWEAT = 51.98; KI = 9.5; VT = 32.9; TTI = 42.8; $CAPE_{uvt} = 0$; $H_{cond} = 3700\,m$ | Same solutions | Translucent plume in the sky |
| 3 | 23 March 2021 $15^{42}$–$14^{56}$, 14 | Clear sky, haze on the horizon | $T_0 = 25.9\,°C$, $f_0 = 36.8\%$, $U_0 = 2$–$3\,m/s$ | LI = 10.49; SWEAT = 64.02; KI = 8.3; VT = 28.7; TTI = 30.4; $CAPE_{uvt} = 0$; $H_{cond} = 3900\,m$ | Same solutions | Translucent plume in the sky |
| 4 | 24 March 2021 $15^{51}$–$15^{58}$, 9 | Clear sky | $T_0 = 29.5\,°C$, $f_0 = 18.8\%$, $U_0 = 3$–$4\,m/s$ | LI = 10.49; SWEAT = 64.002; KI = 8.3; VT = 28.7; TTI = 30.4; $CAPE_{uvt} = 0$; $H_{cond} = 2900\,m$ | Same solutions | Translucent plume in the sky |
| 5 | 25 December 2021 $15^5$–$15^{14}$, 10 | Clear sky over site, high-level St-Cu near site | $T_0 = 15\,°C$, $f_0 = 70\%$, $U_0 = 2$–$4\,m/s$ | LI = 3; SWEAT = 116; KI = 10.5; VT = 3.5; TTI = 37; $CAPE_{uvt} = 15.36$; $H_{cond} = 1300\,m$ | Same solutions | Translucent plume in the sky |
| 6 | 25 December 2021 $16^{10}$–$16^{18}$, 13 | Cu Fr clouds over and near site | $T_0 = 14\,°C$, $f_0 = 68\%$, $U_0 = 2$–$3\,m/s$ | Same | Same solutions | Translucent plume in the sky |
| 7 | 26 December 2021 $14^{59}$–$15^9$, 11 | Clear sky over site, Cu and high-level St-Cu near site | $T_0 = 13.5\,°C$, $f_0 = 69\%$, $U_0 = 3\,m/s$ | Li = 1.5; SWEAT = 118; Ki = 27; VT = 27.3; TTi = 44.6; $CAPE_{uvt} = 0.00$; $H_{cond} = 1200\,m$ | Same solutions | Plume in the sky. Convergence of surrounding clouds toward the jet |
| 8 | 26 December 2021 $15^{36}$–$15^{48}$, 12 | Cu cloud passing over site. The jet is injected directly into cloud | $T_0 = 13\,°C$, $f_0 = 69.7\%$, $U_0 = 2$–$3\,m/s$ | Same | Same solutions | The cloud evaporated as it passed |

**Table 3.** *Cont.*

| No | Date, Time, Duration of Experiment (min) | Weather Conditions | Surface Atmosphere Parameters | Atmospheric Instability Indices | Seeding Materials Injected into Jet | Results |
|----|----|----|----|----|----|----|
| 9 | 29 December 2021 $15^{32}$–$15^{42}$, 10 | St-Cu cloud over site, Cu clouds at North and East | $T_0$ = 13.5 °C, $f_0$ = 68%, $U_0$ = 2–3 m/s | LI = 4.51; SWEAT = 55.01; KI = 18.5; VT = 23.1; TTI = 32.2; $CAPE_{uvt}$ = 5.27; $H_{cond}$ = 1350 m. | Same solutions and 20 kg of powder NaCl/TiO$_2$ | Formation of a gap among the ridge of St-Cu clouds |
| 10 | 30 December 2021 $16^{11}$–$16^{20}$, 9 | Clear sky over site. Cb cloud in the South and over Gulf of Oman | $T_0$ = 12.2 °C, $f_0$ = 73.7%, $U_0$ = 2.7 m/s | LI = 3.81; SWEAT = 205.82; KI = 29.4; VT = 25.5; TTI = 48.9; $CAPE_{uvt}$ = 1.77; $H_{cond}$ = 2200 m | Same solutions and 50 kg powder of CaCl$_2$ and 20 kg of powder NaCl/NiO$_2$ | Translucent plume in the sky |
| 11 | 31 December 2021 $11^{16}$–$11^{32}$, 16 | Low-level St-Cu on site, moving to West. Cb in South and East. Rain on site $11^{55}$-$12^{05}$ | $T_0$ = 11.0 °C, $f_0$ = 88.7%, $U_0$ = 1.8 m/s $J$ = 99 mm/h | LI = 1.07; SWEAT = 229.7; KI = 30.8; VT = 25.5; TTI = 51.3; $CAPE_{uvt}$ = 137.43; $H_{cond}$ = 1600 m | Same solutions and powders of CaCl$_2$ and 20 kg of powder NaCl/NiO$_2$ | Compaction of the seeded cloud volume (visually observed) |
| 12 | 31 December 2021 $15^{55}$–$16^{04}$, 10 | Cu cloud over site, moving to South-West | $T_0$ = 12.3 °C, $f_0$ = 82.0%, $U_0$ = 0.8 m/s | Same | Same solutions and powders | Formation of 15 dBZ radar echo band at $16^{20}$–$16^{25}$ |
| 13 | 3 January 2022 $14^{25}$–$14^{36}$, 11 | Low-level St-Cu cloud on site, moving to East | $T_0$ = 9.0 °C, $f_0$ = 91%, $U_0$ = 4.2 m/s | LI = 3.56.1; SWEAT = 372.9; KI = 36.4; VT = 27.3; TTI = 52.3; $CAPE_{uvt}$ = 449.6; $H_{cond}$ = 1050 m | Same solutions and 20 kg of powder NaCl/NiO$_2$ | Small radar echo in $14^{35}$–$14^{55}$ in the moving direction of the seeded cloud |
| 14 | 5 January 2022 $14^{21}$–$14^{32}$, 11 | Low-level St-Cu cloud on site, moving to East | $T_0$ = 7.7 °C, $f_0$ = 93.8%, $U_0$ = 2.8 m/s | LI = 4.53; SWEAT = 108.09; KI = -14.1; VT = 20.9; TTI = 39.6; $CAPE_{uvt}$ = 12.16; $H_{cond}$ = 1200 m | Same solutions and 50 kg powder of CaCl$_2$ and 20 kg of powder NaCl/NiO$_2$ | Drizzling rain at position and spot of radar echo in the direction of moving of the cloud |
| 15 | 5 January 2022 $15^{25}$–$15^{34}$, 9 | Low-level St-Cu cloud on site, moving to East | $T_0$ = 9.7 °C, $f_0$ = 92% | Same | Same solutions and 20 kg of powder NaCl/NiO$_2$ | Compaction of the seeded cloud volume |
| 16 | 7 January 2022 $10^{50}$–$10^{59}$, 9 | Clear sky over site | $T_0$ = 7.3 °C, $f_0$ = 79%, $U_0$ = 2.5 m/s | LI = 9.49; SWEAT = 76.78; KI = −5.4; VT = 19.7; TTI = 36.1; $CAPE_{uvt}$ = 0; $H_{cond}$ = 1800 m. | Powder of NaCl/NiO$_2$ 20 kg | Formation of a translucent cloud over the jet |
| 17 | 7 January 2022 $12^{19}$–$12^{27}$, 9 | Clear sky over site, small Cu hum near site above 1500 m | $T_0$ = 28.7 °C, $f_0$ = 38%, $U_0$ = 4.5 m/s | Same | Same solutions | Convergence of clouds toward jet |
| 18 | 15 January 2022 $13^{29}$–$13^{43}$, 14 | Clear sky over site | $T_0$ = 29.3 °C, $f_0$ = 33%, $U_0$ = 3.7 m/s | LI = 5.19; SWEAT = 63.41; KI = −19.9; VT = 24.7; TTI = 37.4; $CAPE_{uvt}$ = 0; $H_{cond}$ = 1700 m. | Solutions of CaCl$_2$ and NaCl | Formation of a translucent cloud over the jet |
| 19 | 15 January 2022 $17^{36}$–$18^{03}$, 24 | High-level St-Cu clouds over sky | $T_0$ = 30.5 °C, $f_0$ = 31%, $U_0$ = 4.0 m/s | LI = 6.12; SWEAT = 33.0; KI = −28.5; VT = 27.9; TTI = 16.8; $CAPE_{uvt}$ = 0; $H_{cond}$ = 1830 m. | Solutions of CaCl$_2$, NaCl and powder of CaCl$_2$ 40 kg | A gap in the cloud and its transfer downwind |

**Table 4.** Consumption of solutions and powders per experiment, expected amount of condensed water vapor, and condensation heat.

| Hygroscopic Substance | Solution Flow Rate in Experiments, (g/s) | Substance Mass Flow Rate, $M$ (g/s) | Aerosol Diameter, $d_a$ (μm) | Aerosol Mass, $m_a$ (g) | Aerosol Flow Rate, $N$ (Pcs./s) |
|---|---|---|---|---|---|
| NaCl | 600 | 215.3 | 10 | $1.13 \times 10^{-9}$ | $1.90 \times 10^{11}$ |
| CaCl$_2$ | 1500 | 538 | 10 | $1.13 \times 10^{-9}$ | $4.78 \times 10^{11}$ |
| (NH$_2$)$_2$CO | 800 | 672 | 10 | $6.99 \times 10^{-10}$ | $5.32 \times 10^{11}$ |
| NaCl/TiO$_2$ powder | - | 230 | 5 | $1.41 \times 10^{-10}$ | $1.62 \times 10^{12}$ |
| CaCl$_2$ powder | - | 476 | 50 | $1.41 \times 10^{-7}$ | $3.35 \times 10^{9}$ |

| Hygroscopic Substance | Growth Factor of Mass $k_1 = m_d/m_a$ | Water Vapor Mass Condensed on One Particle, $m_w = k_1 \times m_a$ (g) | Total Mass of Condensed Water Vapor $M_w$ (kg/s) | Latent Heat of Condensation $P_C$ (MW) |
|---|---|---|---|---|
| NaCl | 5.86 | $6.62 \times 10^{-9}$ | 1.26 | 2.89 |
| CaCl$_2$ | 1.67 | $1.87 \times 10^{-9}$ | 0.90 | 2.07 |
| (NH$_2$)$_2$CO | 3.08 | $2.15 \times 10^{-9}$ | 1.15 | 2.63 |
| NaCl/TiO$_2$ powder | 19.4 | $2.74 \times 10^{-9}$ | 4.43 | 10.20 |
| CaCl$_2$ powder | 1.67 | $2.35 \times 10^{-7}$ | 0.79 | 1.81 |

### 4.3. Atmospheric Situations during the Days of Experiments

#### 4.3.1. First Field Campaign

The first campaign took place during the period of 12–28 March 2021, during which weather conditions remained very dry. During this period, there was not a single day when atmospheric conditions met the criteria presented in Figure 5. In the 1 km layer over the site, the air humidity was less than 30%, the convectively retarding layers reached a height of 800–1000 m, and the surface wind speed was 2–5 m/s.

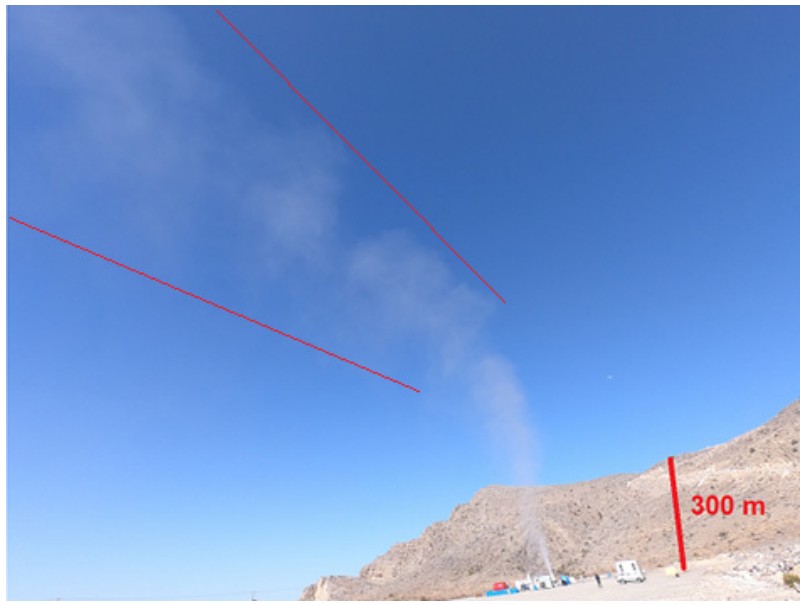

**Figure 5.** A jet in the clear sky on 24 March 2021.

Despite the adverse atmospheric conditions, four trial experiments were carried out (15, 17, 23, and 24 March) to create artificial updrafts in order to study their characteristics. On all these days, the weather in the region was governed by a low-gradient and low-pressure field with low-humidity surface air. Below is a summary of the weather conditions for each trial:

- On 15 March, according to atmospheric sounding data in Abu Dhabi at 12:00 UTC, the presence of high humidity in the 3.5–7 km layer (60–80%) led to the formation of cirrus clouds.
- On 17 March, the surface air humidity was 18–21%. In layers 4.0–4.5; 5.5–6.0; 8.5–10 km, humidity reached 59–71%. However, during the day it was cloudless.
- On 23 March, Meteosat-10 data showed the nearest cloudy zones located over the territory of Iran, in the zone of the low-pressure trough from Transcaucasia through the Caspian Sea. The weather in the region was partly cloudy.
- On 24 March, the humidity of the atmosphere dropped even lower. According to the microwave radiometer observations, the humidity of the surface layer was 20–30%, and 50% at a height of 9–10 km. The nearest cloud fields were also only observed over the territory of Iran.

### 4.3.2. Second Field Campaign

The weather during the second field campaign was more favorable for the creation of artificial clouds. During this period, 15 updraft creating experiments were carried out to initiate the development of artificial clouds under various atmospheric conditions (see Table 2). These experiments were carried out on 25, 26, 29, 30, and 31 December 2021, and 3, 4, 5, and 7 January 2022 on Mount Jebel Jais. On 15 January 2022, two experiments were carried out at a new location—at an NCM site located in the desert area of the UAE.

It follows from Table 2 that from 22 December to 29 December 2021, partly cloudy weather was observed in the Jebel Jais area with the appearance of cumulus humilis (Cu Hum) clouds sometime in the afternoon. Starting from 29 December, heavy rains and thunderstorms were observed over the Gulf of Oman, and with the approach of a cold front from the west, extensive rains and thunderstorms began in the west of the UAE, which covered almost the entire territory of the UAE on the night and morning of 30 December. However, in the afternoon, the area of precipitation decreased significantly, and they shifted to the Gulf of Oman.

From 31 December 2021 to 4 January 2022, several waves of heavy rains with thunderstorms passed over the territory of the UAE, which created flooding in Abu Dhabi and the Jebel Jais gorge. On 5 January, only small pockets of precipitation were observed. From 6 to 15 January, a slight cloudiness of the upper layer prevailed.

## 5. Experimental Results

From Table 3, it follows that 10 experiments were carried out in a cloudless atmosphere over the field campaign site. In four cases, the jet was injected directly into the clouds, including two cases into cumulus clouds and two cases into stratus clouds. The remaining five experiments were carried out in the presence of second-tier clouds of the St-Cu and St type over the site and its environs.

### 5.1. Parameters of Artificial Updrafts

It is not possible to show the full measurements of all parameters of the atmosphere and the jet for each experiment; therefore, we present only generalized results of measurements and observations:

(a) An artificial updraft jet injected into a cloudless space was usually observed against a blue sky as a translucent light plume, which was almost vertical near the ground, and deviated in the direction of the wind as the height increased (see Figure 5).

(b) The temperature of the jet at its edge when entering the atmosphere, according to the Digital Thermometer (TM902C), reached 275–300 °C. It was not possible to measure the temperature within the center of the jet flow.

(c) The speed of the jet at its edge when entering the atmosphere was about 300 m/s. According to the theory of jet flows [52,53], this indicates that the maximum jet velocity on the axis reached 400–450 m/s.

(d) The height of the jet rise, according to the IRTIS-2000C thermograph, reached approximately 600–700 m with a maximum of about 1100–1200 m above the surface (see Figures 6 and 7f).

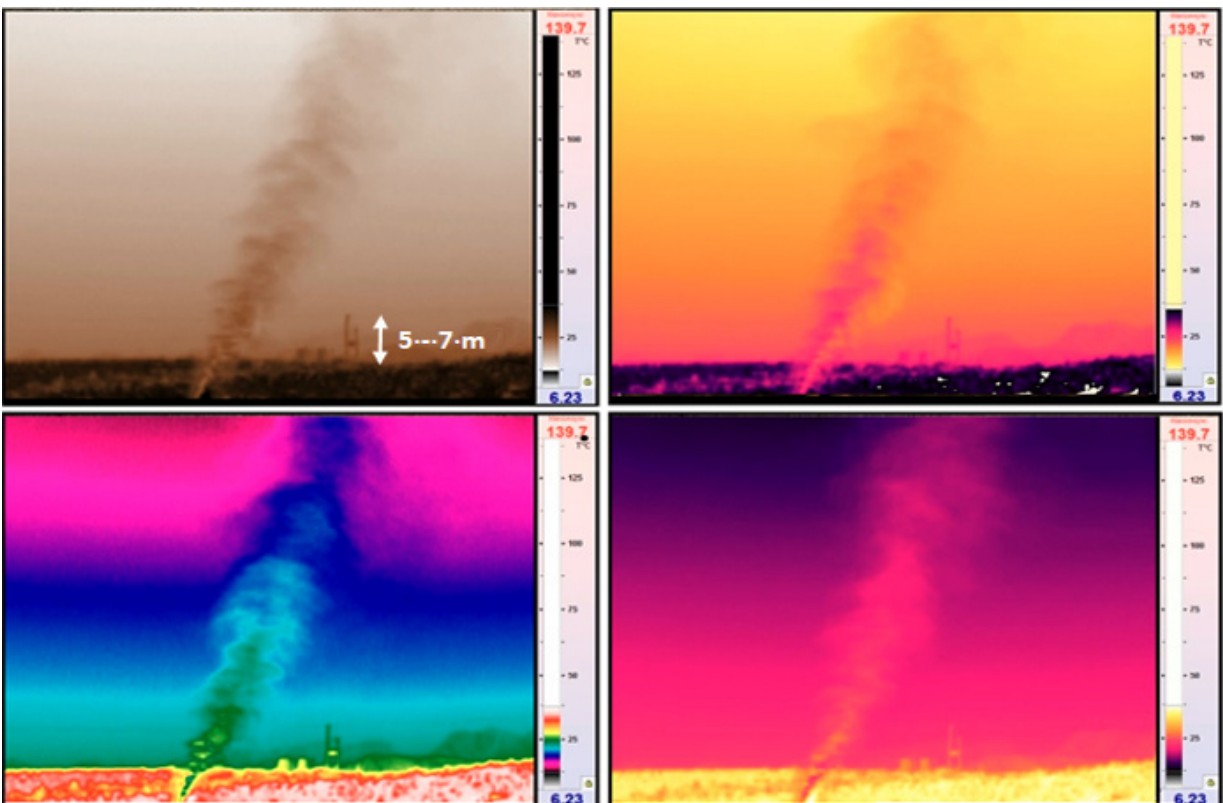

**Figure 6.** Jet form according to thermograph IRTIS-2000C on different visualization regimes, 17 March 2021.

Note: Probably, the jet ascent height was significantly higher in the area where the jet temperature was close to the ambient temperature and the thermograph did not distinguish them from each other. In addition, since the thermograph measurements were made from ground level, our angle did not allow us to qualitatively measure the upper part of the updraft.

(e) The updraft velocity at 400 m was as high as 3 m/s according to the UAV meteorological drone (see Figure 8).

(f) The concentration of natural aerosol over the site in the afternoon in the size ranges of 0.3, 0.5, 1.0, 2.5, 5, 10 μm averaged about 3496, 1235, 335, 71, 18, and 12 in liter, respectively (see Figure 9). This concentration increased up to two times at high wind speed on 16–17 March.

(g) According to the PK200-03 pyranometer, the solar flux on a flat surface varied on different days within the range of 0.85–1.0 kW/m$^2$.

(h) The noise level of the experimental setup, according to noise meter MEGEON 92131, reached 132 dB of sound-pressure level with a maximum of 135 dB at maximum engine speed.

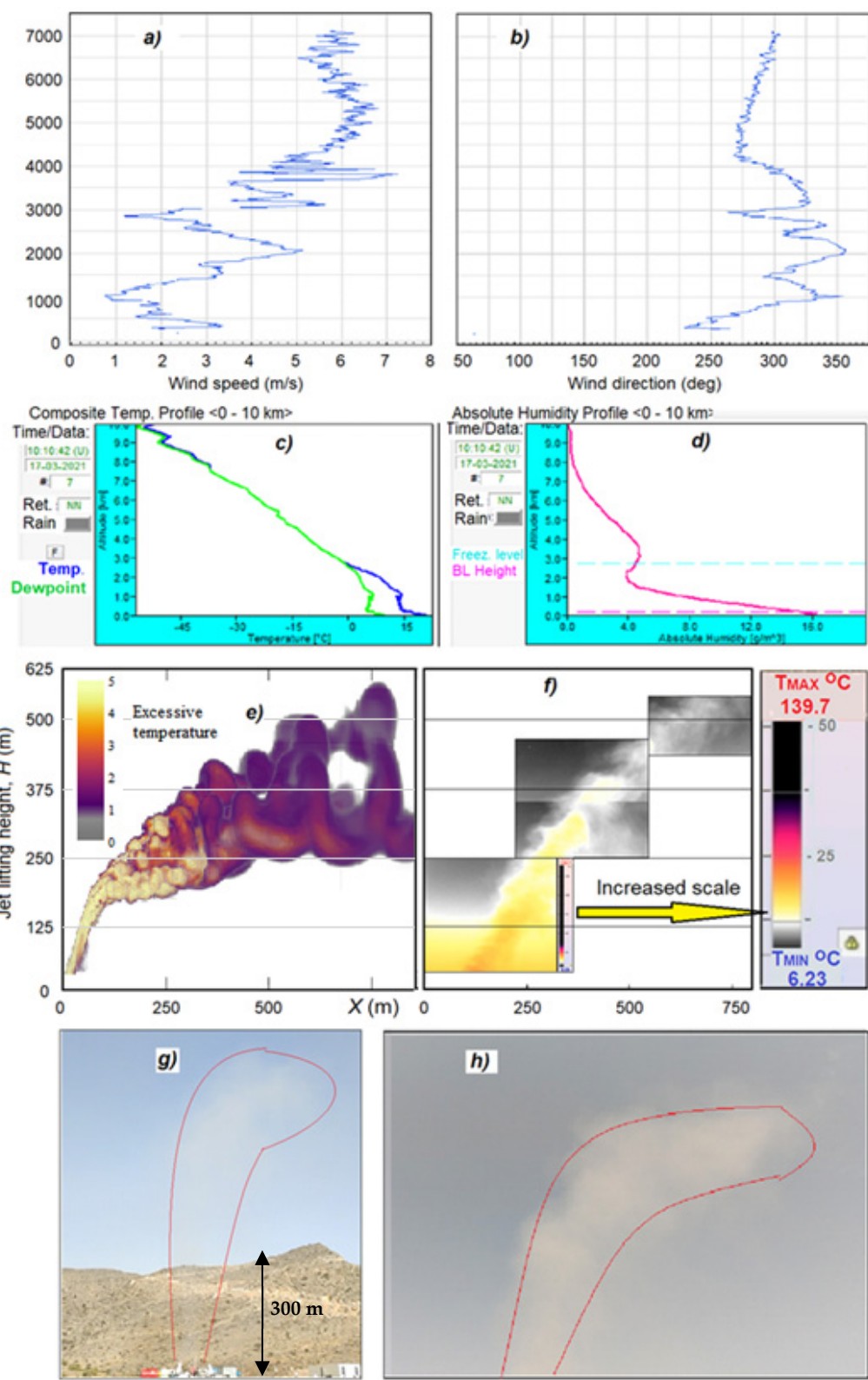

**Figure 7.** Comparison of the simulated and observed jet characteristics, initiated at an altitude of 1595 m above sea level during trial field experiment on Jebel Jais Mountain of UAE on March 17, 2021. Wind speed (**a**) and direction (**b**) measured by Wind Lidar Halo Photonics Streamline XR; temperature (**c**) and specific humidity (**d**) profiles measured by microwave radiometer RPG HATPRO-G4 PRO; simulated jet shape (**e**); combined thermal images of jet (**f**) measured by infrared scanning thermograph IRTIS-2000C at 3 viewing angles; visual images by GoPro Hero-9 Camera (**g,h**).

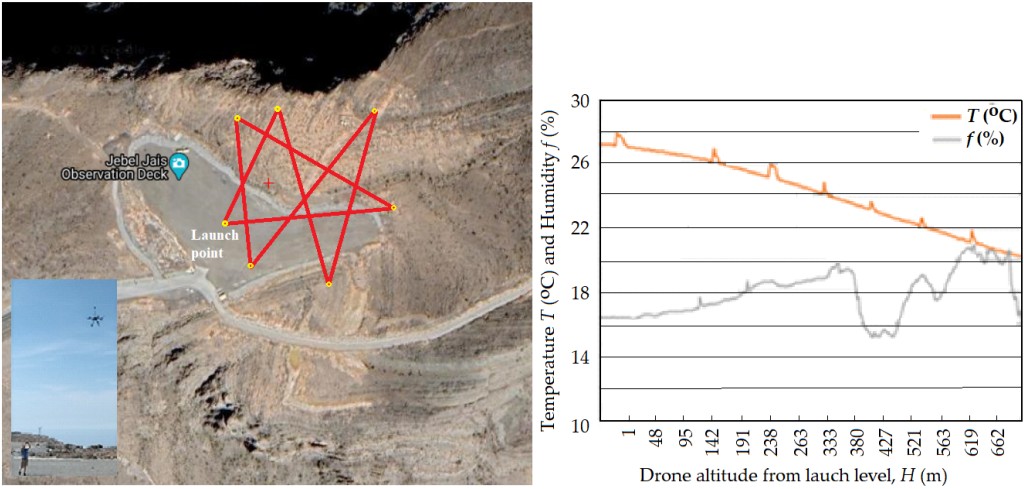

**Figure 8.** Typical weather drone trajectory over the experimental site and measurements of temperature and humidity in artificial updraft and environment.

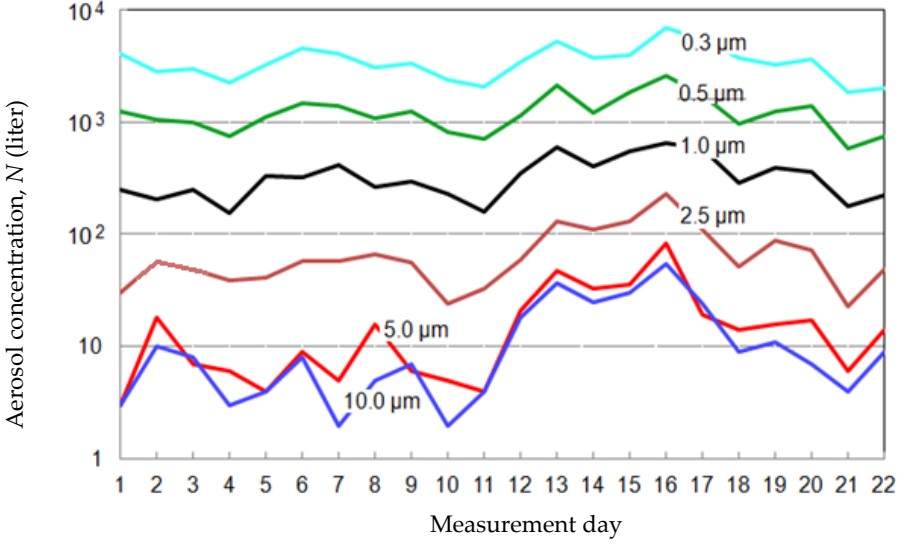

**Figure 9.** Concentration of natural aerosol over the site in the afternoon in the size ranges of 0.3, 0.5, 1.0, 2.5, 5, 10 μm.

Figure 8 shows an example of a meteorological drone flight path (in red) used to measure jet parameters and the surrounding atmosphere, including temperature, humidity, vertical and horizontal velocities, aerosol concentration in five size ranges, and GPS coordinates.

Hygroscopic aerosols are introduced into the jet to stimulate the condensation of water vapor and feed the energy of the jet with the heat of condensation. From Table 2, it follows that the use of NaCl aerosols in this respect is more efficient than carbamide and $CaCl_2$ aerosols. However, the low hygroscopic point of $CaCl_2$ (6%) can provide condensation heat at low air humidity. The use of $NaCl/TiO_2$ aerosol was expected to be very effective, on which many times more water vapor can condense, starting from an air humidity of 52% [42,43].

*5.2. Effects Discovered as a Result of Experiments*

**Effect 1:** Figure 10 shows an example of low-level cloud convergence. It is clearly seen that this happens in a short time. In just 2.5 min, the window of clear sky over the position began to decrease due to the displacement of the surrounding clouds in the direction of the jet. However, it is possible that this was not due to the action of the jet, but due to the convergence over mount Jebel Jais of the winds blowing from the gulfs on the left and right.

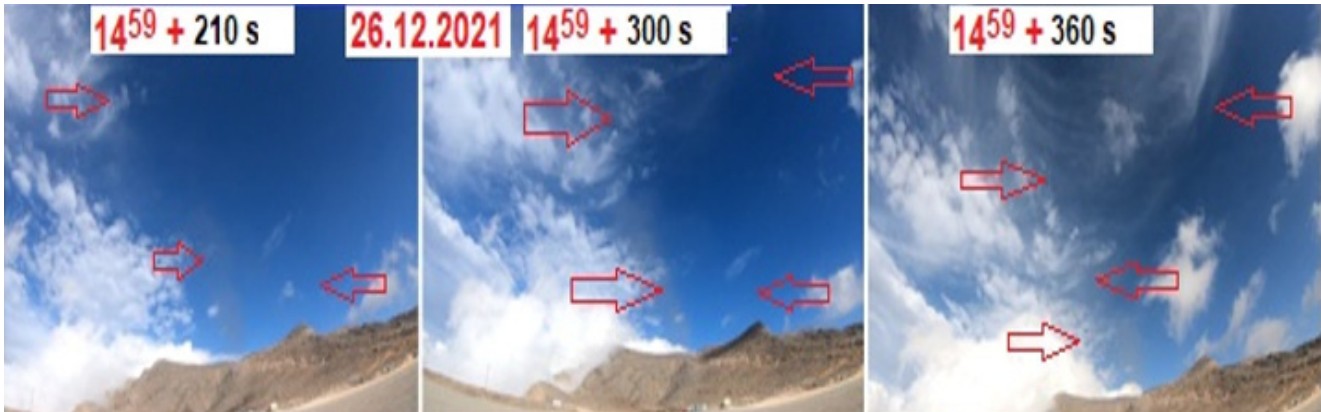

**Figure 10.** Convergence of clouds in the direction of the jet. The engine was turned on at 14:59 local time, leap seconds show the time elapsed since the start of experiment.

**Effect 2:** In a clear sky above a jet saturated with hygroscopic aerosols, translucent aerosol clouds usually form. An example of them is shown in Figure 11. These clouds are assumed to be accumulations of aerosols and water droplets formed as a result of the condensation of water vapor on $CaCl_2$ aerosols, which have a hygroscopic point at 6% humidity. Water vapor condensation on NaCl and $(NH_2)_2CO$ aerosols does not occur due to low air humidity.

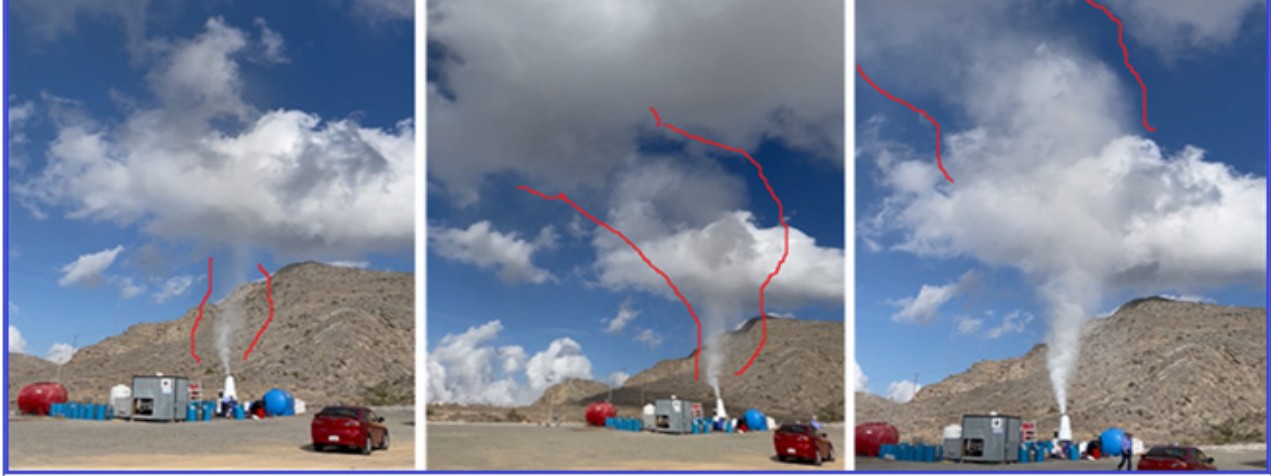

**Figure 11.** Transparent aerosol-droplet clouds created in a partially cloudy sky on 28 December 2021.

**Effect 3:** Figure 12 shows the injection of a jet into a cloud moving through the field site on 31 December (the same case was on 5 January 2022). In these cases, only some compaction and changes in the visual structure (increased visibility tunnels) of the clouds are noted.

**Effect 4:** One case was noted when a shallow cloud moving over a jet saturated with $CaCl_2$, NaCl, and $(NO_2)_2CO$ solutions partially disappeared. As the cloud passed over the jet, an increase in the level of the cloud base and its gradual disappearance was noted (Figure 13). The cloud volume reduction lasted about 4 min. It can be assumed that the partial disappearance of this cloud can be associated with the enlargement of cloud droplets and drizzle, as well as the evaporation of the remaining part of the droplets under the action of a hot jet. Subsequently, this cloud was not restored to its initial shape.

**Effect 5.** In three experiments, when clouds passed over the jet, according to the data of the NCM radar network, the appearance of a radar echo on the leeward side was detected (Figure 14). These radar echoes with a reflectivity of no more than 20 dBZ corresponded to a zone of precipitation with an intensity of less than 1 mm/h. They appeared 10–15 min after

the start of the jet-producing device at a distance of about 3–5 km from the experimental site. The radar echo spots had a transverse size of about 1.5–2.0 km and an existence time of about 10–15 min. It is quite plausible that the formation of these radar echoes is due to the fact that hygroscopic aerosols introduced into the jet contributed to the coarsening of cloud particles and the formation of precipitation. However, there is an alternative explanation: the appearance of these radar echoes may be due to the refraction of radar waves and incomplete suppression of interference from ground clutters. The distance from site to the nearest weather radar was about 130 km.

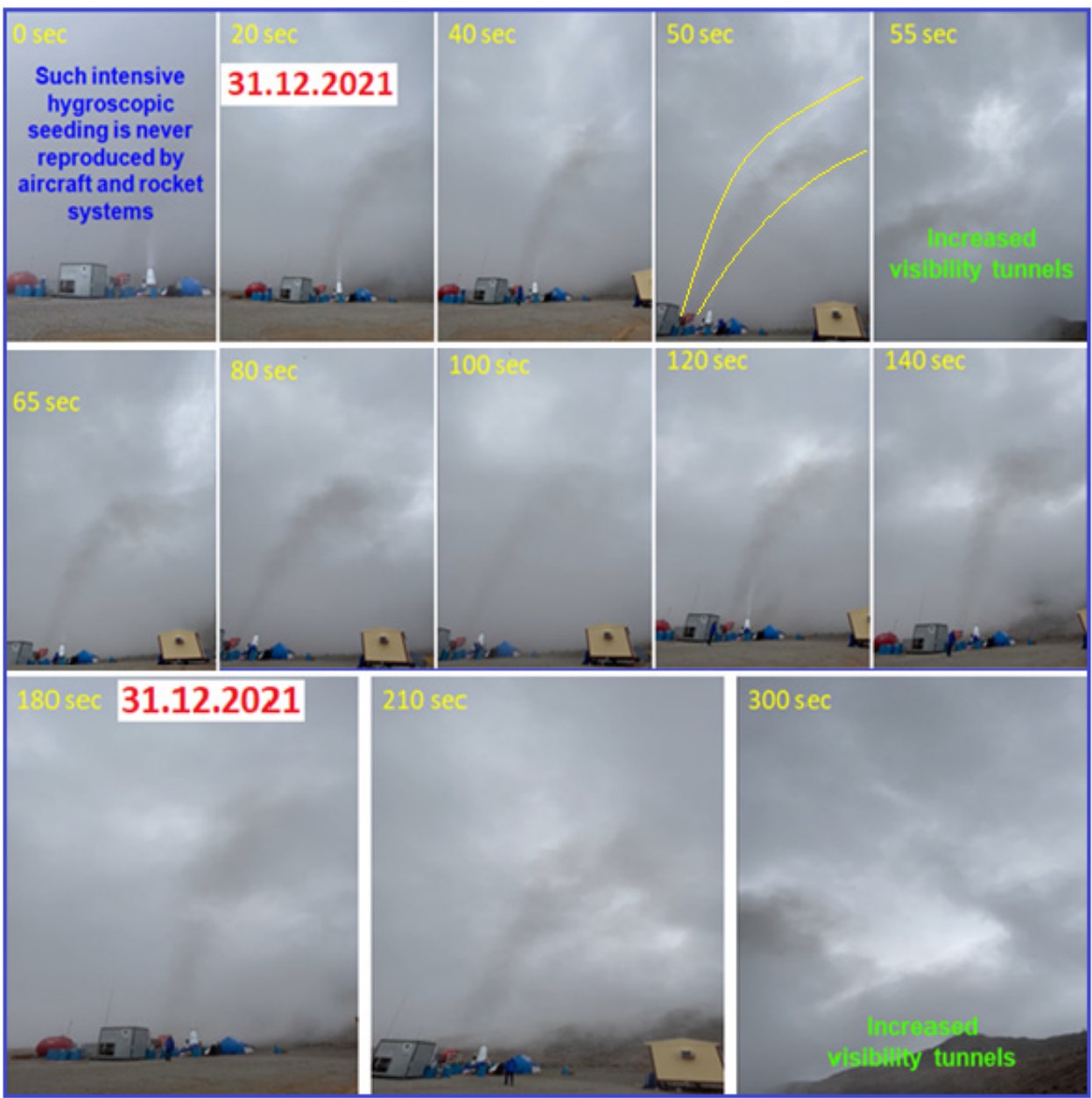

**Figure 12.** Impact on Cu cloud approaching experimental site from East. Rain and thunderstorm over site from 11:55 to 12:05 local time with a break. Rain intensity $I$ = 99 mm/h, peak $I_{max}$ = 125 mm/h, which caused the water flow in the valley next to the experimental site.

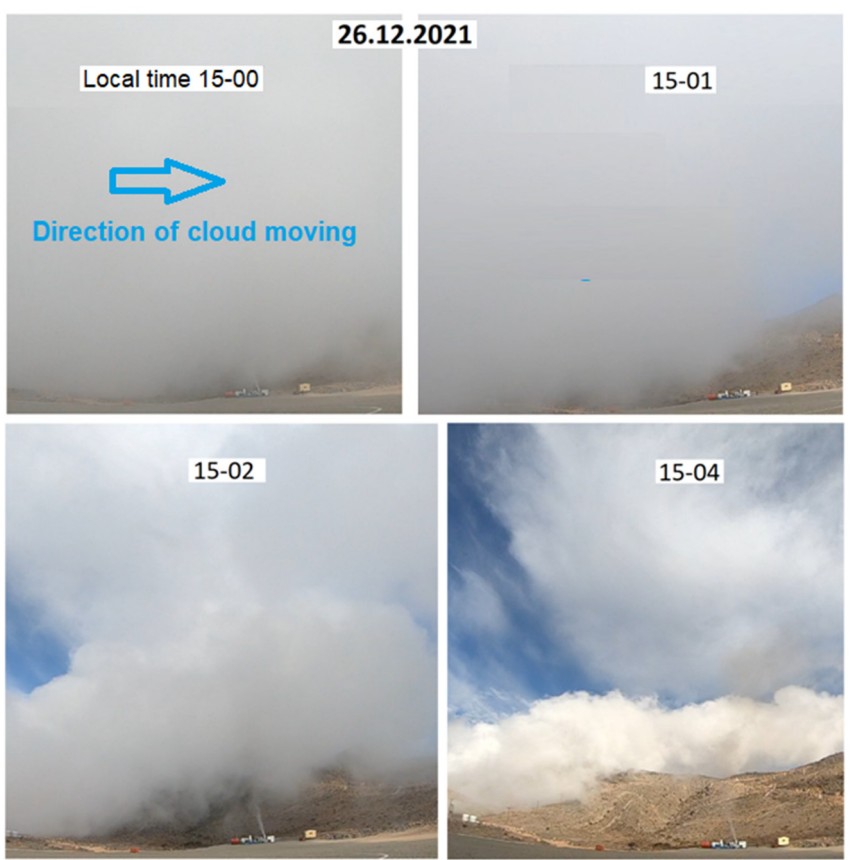

**Figure 13.** Example of partial cloud disappearance when passing over a hot jet saturated with solutions of hygroscopic substances.

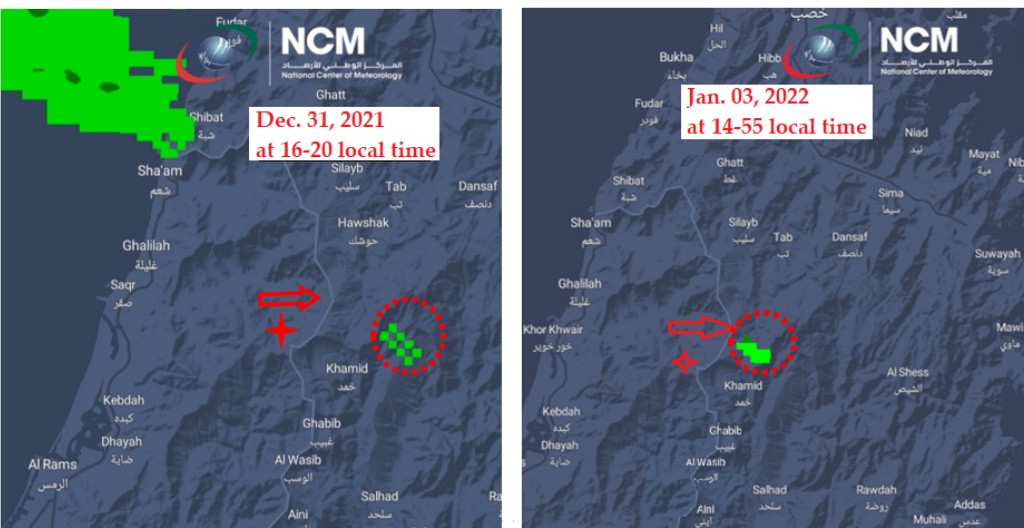

**Figure 14.** Radar maps with radar echo of artificial rain at 16–20 of local time, 31 December 2021, and 14–55, 3 January 2022. Here, red arrow—wind direction, hollow cross—field site location, and dotted circle—radar echo signature.

## 6. Discussion of Experimental Results

During the two field campaigns, 19 full-scale real atmosphere experiments on the creation of artificial clouds at different atmospheric conditions were carried out on Mount Jebel Jais in the UAE. A customized system was deployed to generate updrafts saturated with different hygroscopic aerosol. A total of 10 experiments were conducted in a cloudless

sky; two with cumulus clouds passing over the site; two with stratocumulus clouds passing over the site; five in the presence of stratocumulus and stratus clouds over the site and in the ambient environment, from which three cases of second-tier clouds (at an altitude of more than 3000 m) and two cases of lower-tier clouds (near the experimental site level). As a result of these experiments, the following physical effects were noted:

- 13 cases with a cloudless sky—the formation of a jet plume in the form of almost transparent aerosol cloud;
- three cases—compaction and further formation of the enlightenment zone of clouds into which a jet saturated with hygroscopic aerosols was injected;
- three cases—the appearance of a radar echo spot on the downwind side 15–20 min after the start of the jet-producing device. Perhaps this was due to the action of the jet directly on clouds like Cu Hum (two cases) and St (one case);
- two cases—convergence of small clouds (Cu Hum type) in the direction of the jet (but this may be due to wind convergence);
- one case—scattering of a cumulus cloud as it passed over the jet.

No experiment succeeded in creating a deep convection with development of artificial convective cloud in a clear sky. Strengthening of natural clouds and stimulation of local precipitation formation obtained in some of the experiments also has no practical significance. These results were significantly lower than expected based on theoretical simulations [36–39]. The reasons for such modest results are that the condensation of water vapor on the used hygroscopic aerosols and the release of condensation heat do not occur during the rise of the jet, but occur later or not at all. Therefore, the artificial upward flow created by the jet engine did not receive energy replenishment due to the heat of condensation. The jet rose in the atmosphere under the influence of its initial energy. However, such a jet, according to simulations data, cannot reach the level of natural condensation and start the cloud formation mechanism even at wind speed $U = 1 + 0.005\,h$ and temperature lapse rate $\gamma = 8.5\ °C/km$ [36].

During the experiments, it was noticed that a surface wind with a speed of more than 5 m/s deflects jets of fog cannons, but cannot deflect the jet of a turbojet engine with an initial speed of more than 300 m/s. However, at a height of 200–300 m/s, the jet velocity decreases to 4–6 m/s, and the wind, which has a comparable speed, tilts the jet and limits its rise.

On days with experiments, the height of the condensation level was from 1100 to 2200 m. At the Jebel Jais site, for 6 days, the level of condensation was below the jet start level. However, the low specific humidity at this altitude and insufficient relative humidity limited the condensation of water vapor to form clouds.

It should be noted that, unlike a turbojet engine jet penetrating a certain layer of the atmosphere like a needle, natural thermals that ensure the development of convective clouds over the solar slopes of mountains have a large base area. Although the thermal temperature excess over ambient air ($\Delta T$) and the rate of their rise are much less than that of a vertically directed jet of a turbojet engine, their energy is sufficient for the development of thermal convection and the formation of clouds (Figure 15). This indicates the advisability of creating artificial ascending streams with a large base area.

It should also be noted that 17 out of 19 experiments were carried out in the spring and winter field campaigns at an altitude of 1600 m, which in most cases was above the level of condensation and the base of the lower-tier clouds, but lower than the second-tier clouds. The low relative and absolute humidity at this height precluded the condensation of water vapor and the replenishment of the jet with the heat of condensation. In addition, without such replenishment, the jet energy was not sufficient to overcome temperature inversion in the near-ground atmosphere, the destructive effect of the wind, and the creation of precipitation-forming convective clouds.

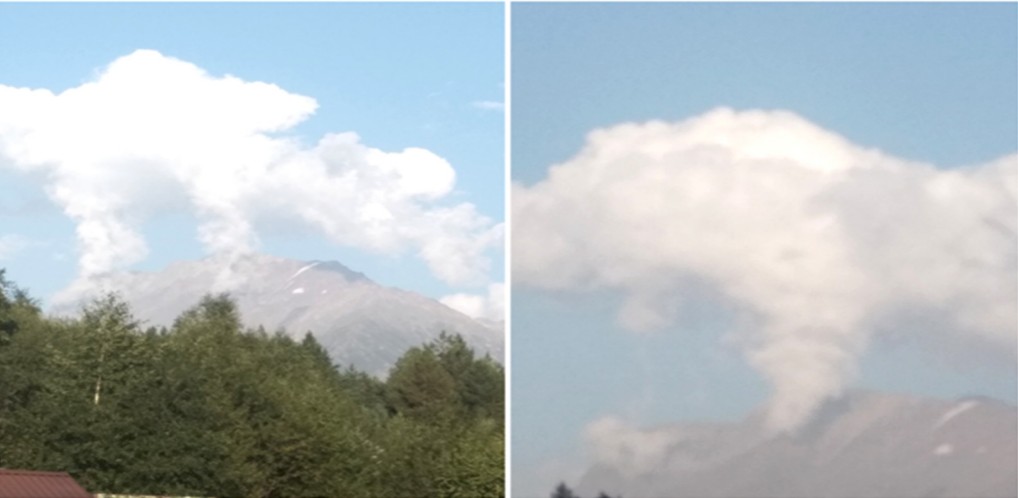

**Figure 15.** Example of convective clouds formed in the afternoon over the sunny mountain slope. Terskol, Northern Caucasus, Russia. 28 August 2021, time 17³⁶.

## 7. Conclusions and Recommendation

The field experiments described in this paper envisaged the initiation of convective cloud development through artificial updrafts created by a vertically directed jet saturated with different hygroscopic aerosols. The jet introduction into a cloudless sky led to the creation of translucent aerosol-droplet clouds. Jet introduction into existing clouds led to the effective absorption of cloud water content along the jet propagation path and enhancing of in-cloud visibility, finally producing local zones of weak precipitation on the downwind side. In some cases, the evaporation of clouds passing over the hot jet was also observed.

The reason for the insufficient effectiveness of the method is that the low air humidity excluded the condensation of water vapor on the types of hygroscopic aerosols used, and also that the condensation process is rather slow. Therefore, the jet in the process of ascent does not receive an energy feeding at the expense of condensation heat. The lack of recharge deprives the tested method of an advantage over previously proposed methods of creating artificial clouds using meteotrons [14,29,31,50]. As a result, the jet power of the aircraft engine used is insufficient to overcome the inversion layers and initiate the development of convective clouds.

To enhance the efficiency of the proposed method, it is recommended to:

- Create an experimental device consisting of several and more powerful aircraft engines arranged to create a powerful updraft over an area comparable to the size of natural thermals.
- Develop a new hygroscopic aerosol that can provide: faster condensation of water vapor when the jet rises (i.e., in 5–10 s); water vapor condensation at air humidity less than $f_o < 40\%$; and condensation of water vapor, many times its mass, for example, by further improving the core/shell $NaCl/TiO_2$ aerosol.

Increasing the speed of the used hygroscopic agents (as well as glaciogenic agents) is no less important for the practice of weather modification than the optimization of the temperature and humidity of the air at which they operate. The speed and temperature–humidity regime of seeding materials are some of the most important parameters affecting the efficiency of cloud seeding.

Numerical simulations of the jet produced by one, two, four, eight, and sixteen engines of the D-30 type, confirm that the volume of the jet and the height of its rise increase with an increase in the number (or power) of jet engines (Figure 16). An increase in the number (power) of engines from one to 16 leads to an increase in the height of the jet by a factor of three and an increase in the volume of the jet by dozens of times. The horizontal jet length in the direction of the wind also increases significantly (from 1.1 km to 2.5 km).

These effects increase as the temperature lapse rate increases. At $\gamma$ = 9.5 °C/km, the jet, overcoming wind transfer, rises to a height of more than 4.5 km and stretches horizontally up to 6 km, and the volume of the jet reaches 15 km$^3$, which is comparable to the volume of Cu Cong or Cumulonimbus clouds.

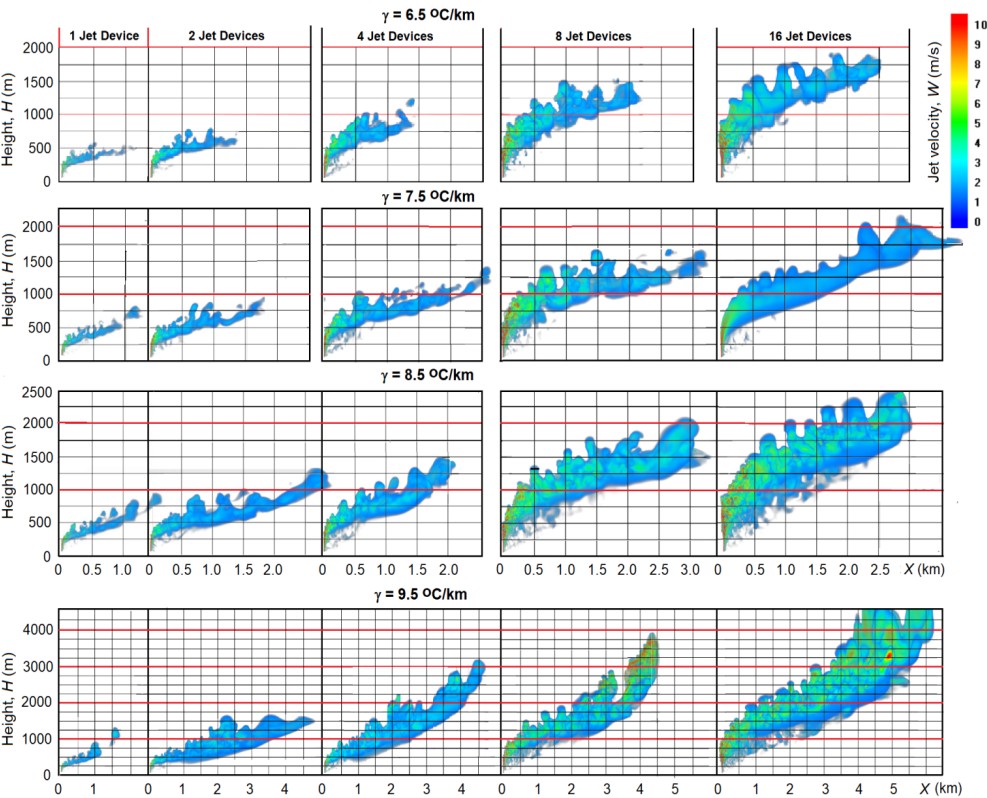

**Figure 16.** Vertical cross sections of the jet velocity field $W$ (m/s) created by an experimental device consisting of 1, 2, 4, 8, and 16 turbojet engines D-30. Atmospheric conditions: vertical wind profile $U$ = 1 + 0.005 $h$ (m/s); temperature lapse rate $\gamma$ = 6.5, 7.5, 8.5 and 9.5 °C/km, no inversion layers.

It follows from Figure 17 that the height of the jet rise depends on the power of the turbojet engine (or the number of engines) to the power of 0.4, and not to the power of 0.25, as Vulfson and Levin [29] supposed.

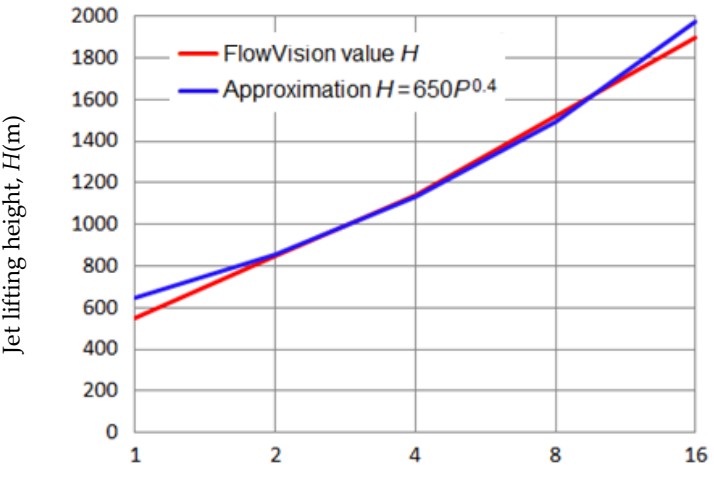

Multiplicity of jet engine power $P$ increase

**Figure 17.** Dependence of jet lifting height $H$ from turbojet engine power $P$.

The creation of an experimental device containing 10 more powerful aircraft engines (such as PD-35 type) will make it possible to achieve a power of about 3.5 GW. This is three to six times more powerful than the Dessens, Vulfson and Levin, Kuznetsov, and Konopasov meteotrons. According to simulation data, such a device generates thermal convection that can stimulate the development of convective clouds in favorable atmospheric conditions. The introduction of even ordinary hygroscopic NaCl aerosol or novel core/shell $NaCl/TiO_2$ aerosol into such a powerful jet would intensify precipitation formation.

Finally, new field experiments on the creation of artificial clouds in the UAE are recommended to be carried out at an altitude of no more than 500 m above sea level, where the specific air humidity is much higher.

**Author Contributions:** Conceptualization, overall project supervision, data analysis, and writing original draft, M.T.A.; organization of field experiments, equipment preparation, and experimental data analysis, A.M.A.; adaptation of mathematical model and conceptualization, A.A.A.; simulations on supercomputer and data analysis, J.V.F.; supervision of simulations and organization of supercomputing facilities, A.E.S.; funding acquisition and formal analysis, A.A.M.; manuscript review, O.A.Y.; proofreading and manuscript review, Y.W.; resources and formal analysis, E.S., D.A.S. and S.E. All authors have read and agreed to the published version of the manuscript.

**Funding:** This work was supported by the National Center of Meteorology, Abu Dhabi, UAE under the UAE Research Program for Rain Enhancement Science (UAEREP, grant No APP-REP-2017-02120).

**Institutional Review Board Statement:** Not applicable.

**Informed Consent Statement:** Not applicable.

**Data Availability Statement:** All data generated or analyzed during this study are included in this published article.

**Conflicts of Interest:** The authors declare no conflict of interest.

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
