# Peer review of "Results of Field Experiments for the Creation of Artificial Updrafts and Clouds"

_atmosphere, doi:10.3390/atmos14010136_

Round 1
Reviewer 1 Report (Previous Reviewer 1)
I have no further suggestions.
Author Response
We thank the reviewer for his time devoted to our article and valuable comments. This made the article better.
Reviewer 2 Report (Previous Reviewer 2)
Citation styles in this manuscript are not uniform.
There are places where there is a sudden line break (e.g., L238).
The table caption is incorrect (e.g., L490).
In addition, the authors should carefully double-check the entire manuscript.
Author Response
Q1. Citation styles in this manuscript are not uniform.
A1. Corrections are made, thank you for remark.
Q2. There are places where there is a sudden line break (e.g., L238).
A2. Corrected, thank you for this comment.
Q3. The table caption is incorrect (e.g., L490).
A3. Line 492 – Phrase “Table 3. Continued” was changed for “Table 4. Continued”
Q4. In addition, the authors should carefully double-check the entire manuscript.
A4. Manuscript check completed, flaws found fixed.
We thank the reviewer for his time devoted to our article and valuable comments. This made the article better.
Reviewer 3 Report (Previous Reviewer 3)
The authors have addressed my concerns satisfactorily.
Author Response
We thank the reviewer for his time devoted to our article and valuable comments. This made the article better.
This manuscript is a resubmission of an earlier submission. The following is a list of the peer review reports and author responses from that submission.
Round 1
Reviewer 1 Report
Comments:
(1) The manuscript is generally well written, the methods and results are also clearly explained. On the other hand, while the introduction explains prior work well, there's not enough explanation of the current manuscript: what you will be doing, sections of the paper, etc. This should be improved.
(2) Some of the subscripts and notation need reformatting: e.g. line 220, etc.
Author Response
Q1. The manuscript is generally well written, the methods and results are also clearly explained. On the other hand, while the introduction explains prior work well, there's not enough explanation of the current manuscript: what you will be doing, sections of the paper, etc. This should be improved.
A1. At the end of the Introduction section, a brief indication of the contents of all sections of the article has been added.
Q2. Some of the subscripts and notation need reformatting: e.g. line 220, etc.
A2. Text correction completed, the problem was related to a change in font type, in which the Greek character γ was replaced with an incorrect character.
Authors express their gratitude to the reviewer remarks. Their correction led to the improvement of the manuscript.
Reviewer 2 Report
In this manuscript, the authors investigated the creation of artificial updrafts and clouds using the jet from an aircraft engine. Investigation of the creation of artificial clouds and precipitation is very important. However, I think the current form of this manuscript is not acceptable to publish and I request you change several parts of this manuscript. Detailed comments are followed.
First of all, the manuscript is not well organized. This is true of the entire manuscript, but especially in the introduction and section 4. Poorly organized manuscript makes the reader very uncomfortable to read. The format of the graph is also inconsistent and needs to be redrawn.
What is a hygroscopic point and how is it calculated in this manuscript? In practice, not only the composition of the particles, but also the size seems to matter, how is this taken into account?
In eq. 1, the formula is incomplete, and the description of the formula is also lacking. In addition, there are many typos and misspelled numbers. Authors should check carefully.
The fonts in the tables are too large and difficult to see because they are not well organized.
According to the manuscript, Pc=153. What does this value mean? What is the x-axis in Figure 1.
Fog cannons JY-60 and WP-60 were used in this study. What is the difference between the two models?
It is better to organize Table 1 and Figure 4 so that they correspond to each other.
It is recommended to modify the color of the left side and the flow chart of Figure 5 to be consistent.
According to Figure 5, the experiment is to be performed only when U0 < 2 m/s. However, in most cases in Table 1, U0 is greater than 2 m/s. Why?
Section 5: Is there a difference in aerosol concentration before and after the experiment? And NaCl/TiO2 particles were also used in this experiment, and the explanation for this result seems to be insufficient. The author proposed the development of new material. But isn't it already used with the NaCl/TiO2 experiment in this study?
Line 586: The authors mentioned that aerosols and water droplets formed as a result of the condensation of water vapor on CaCl2 aerosols, not NaCl and (NH2)2CO. How do we know this?
Have any model experiments been performed under the same weather conditions as the experimental cases in this study? It seems that the model results like this should be used as data to supplement the experimental results.
Author Response
Q1. First of all, the manuscript is not well organized. This is true of the entire manuscript, but especially in the introduction and section 4. Poorly organized manuscript makes the reader very uncomfortable to read.
A1. In the introduction section, we have added a description of the content of the article sections. Section 4 includes a brief description of the experimental data, the methodology for choosing the suitable meteorological situation for the experiment, the procedure for conducting instrumental measurements, a summary table for each day with experiments and the consumption of hygroscopic substances of each type. To improve visual perception, Figure 5 was modified, the font of the tables was changed.
Q2. The format of the graph is also inconsistent and needs to be redrawn.
A2. The comment is accepted. Figures 1, 5, 9, 10, 11, 20 have been visually improved.
Q3. What is a hygroscopic point and how is it calculated in this manuscript? In practice, not only the composition of the particles, but also the size seems to matter, how is this taken into account?
A3. A hygroscopic solid material can be identified through its hygroscopic point and water uptake capacity. The hygroscopic point (hgp) represents the threshold value of the relative humidity in the air above which the solid substance starts adsorbing water vapor (Tereshchenko, 2015). It describes the relationship at equilibrium between the water vapor pressure (Psol) and its surrounding environment with respect to the partial pressure of the water vapor in the air (PH2O) at a specific temperature. hgp is generated as a percentage given by
hgp = 100% × (Psol / PH2O).
A.G. Tereshchenko, “Deliquescence: hygroscopicity of water-soluble crystalline solids,” Journal of Pharmaceutical Sciences, vol. 104, no. 11, pp. 3639–3652, 2015.
The hygroscopic point depends on the properties of the material, the particle size, air temperature and pressure. Adequate consideration of all possible factors is a difficult task. In this work, we simplistically assume the values of hygroscopic points hgp1 < 6% (CaCl2), 41 < hgp2 < 70% (Carbamide), and 71 < hgp3 < 80% (NaCl), regardless of the particle size (which in our device are formed by the size 5 - 15 microns) and temperature. As soon as the air humidity exceeds these values, we assume that the process of water vapor condensation begins.
The corresponding definition of the hygroscopic point was included in the text of the article at the first mention of the concept of the hygroscopic point.
Q4. In eq. 1, the formula is incomplete, and the description of the formula is also lacking. In addition, there are many typos and misspelled numbers. Authors should check carefully.
A4. Eq. 1 and its description have been supplemented; typos and misspelled numbers have been corrected. This and other similar errors in the text are related to the incorrect display of Greek characters, which appeared probably as a result of changing the font.
Q5. The fonts in the tables are too large and difficult to see because they are not well organized.
A5. The font size of the tables has been reduced.
Q6. According to the manuscript, Pc=153. What does this value mean? What is the x-axis in Figure 1.
A6. Pc is the power released at condensation of water vapor on N = 2·1011 sec-1 of hygroscopic NaCl/TiO2 micro powder particles with diameter of 10 μm and total mass of about 230 g/s released into the jet
Pc = M'wt × q = 66.8×2260 » 153 MJ/s = 153 MWt.
It’s description is given in more details in the paper -
Abshaev, M.T., Abshaev, A.M., Aksenov, A.A., Fisher, I.V., Shchelyaev, A.E., Mandous, A., Wehbe, Y., El-Khazali., R,CFD simulation of updrafts initiated by a vertically directed jet fed by the heat of water vapor condensation. Sci Rep 12, 9356 (2022). https://doi.org/10.1038/s41598-022-13185-2Al Mazroui, A. (2017). Advancing the science, technology and implementation of rain enhancement. Project of water security solutions in arid and semi-arid regions and beyond. DOI: http://dx.doi.org/10.13140/RG.2.2.34696.21762.
Axis x in Fig. 1 denotes to the value of feeding energy related to Pc.
Q7. Fog cannons JY-60 and WP-60 were used in this study. What is the difference between the two models?
A7. The characteristics of these two fog cannons are relatively similar in terms of the flow rate of the working fluid per unit time and the length of the spray plume. The greatest difference is the formed spectrum of drops. The JY-60 cannon produces droplets in the 10-20 micron range, while the WP-60 cannon produces droplets with a modal size of 15 microns. These cannons were used to simultaneously spray different hygroscopic substances.
Q8. It is better to organize Table 1 and Figure 4 so that they correspond to each other.
A8. The proposal is interesting, but unfortunately, in the table 1, unlike in Figure 4, there are 2 more components - Meteorological satellites Meteosat-10 and Abu-Dhabi airport air sounding data and some information about the measured parameters.
Q9. It is recommended to modify the color of the left side and the flow chart of Figure 5 to be consistent.
A9. These changes have been made.
Q10. According to Figure 5, the experiment is to be performed only when U0 < 2 m/s. However, in most cases in Table 1, U0 is greater than 2 m/s. Why?
A10. In the course of two field campaigns, we practically did not manage to find such an ideal situation in terms of wind calmness, which, of course, affected the results of the experiments, but still it is a scientific result.
Q11. Section 5: Is there a difference in aerosol concentration before and after the experiment?
A11. Our measurements of the aerosol were carried out near the ground at a height of 2 m before experiments. The aerosol created by the action of the experimental machine was carried up with the jet and to the leeward side. In this regard, we did not measure the aerosol during and after the experiments.
Q12. NaCl/TiO2 particles were also used in this experiment, and the explanation for this result seems to be insufficient.
A12. The preliminary current conclusion is that the condensation of vapor on the NaCl/TiO2 particles substance begins with some delay of 20 - 40 seconds. During this time, the jet does not have time to get enough recharge. Research in this direction is ongoing, and new results may be published in subsequent articles.
Q13. The author proposed the development of new material. But isn't it already used with the NaCl/TiO2 experiment in this study?
A13. As stated in the answer to the previous question, water droplets form on NaCl/TiO2 particles with a delay. In a new hygroscopic material, if it can be obtained, the rate of droplet activation at which the heat of condensation is released can be accelerated.
Q14. Line 586: The authors mentioned that aerosols and water droplets formed as a result of the condensation of water vapor on CaCl2 aerosols, not NaCl and (NH2)2CO. How do we know this?
A14. According to atmospheric radio sounding data on December 28, 2021 (website of the University of Wyoming), in a layer of 500 m from the ground at the field site on Jebel Jais Mount, the air humidity varied from 30 to 60%. It is known that the activation of water droplets on micron size salt particles such (NaCl and its combinations) and particles of urea (NH2)2CO) usually does not occur. At the same time, the CaCl2 aerosol can contribute to the formation of droplets at such low humidity (starting from 6% humidity).
Q15. Have any model experiments been performed under the same weather conditions as the experimental cases in this study? It seems that the model results like this should be used as data to supplement the experimental results.
A15. Fig. 8 represents one preliminary comparison of the simulated and observed jet characteristics, initiated on March 17, 2021. We got a good agreement for the jet shape and height while other characteristics are under investigation. At the moment, we are modeling about ten cases with experiments and the next publication will be devoted to comparing the model and measurements.
Authors express their gratitude to the reviewer for fair remarks. Their correction led to the improvement of the manuscript.
Reviewer 3 Report
This study puts to test the new proposed methodology of creating rain clouds by a combination of hot air jet and hygroscopic seeding with materials that deliquesce at low RH for releasing extra latent heat and creating extra cloud water and lifting.
Although the experiments did not achieve what was hoped for, it is very important to publish the outcome, because it addresses experimentally several hypotheses that have been highly publicized and gained some traction.
However, before it can be published, the paper has to undergo a major revision to account more fully, quantitatively and objectively, for the experimental outcomes, and to clarify several important quantitative questions with respect to the underlying hypothesis.
Specific comments:
Line 69: The statement is incorrect. For example, numerical weather predictions of the NMC can definitely predict isolated air mass convection over the mountains.
Lines 154-156: Please provide references for the trials to increase rainfall by creating heat islands.
Lines 196-198: When describing the methodology, please define what are the h points. Then quantify the amount of condensed water and released latent heat at the deliquescence point for 1 g of the different types of hygroscopic materials. This is a missing fundamental quantity.
Line 262-263: The value of 295 g of absorbed water per 1 g of NaCl/TiO2 is fantastic. Is it realized already at the deliquescence point or only at the point of RH=100%?
I had to go to the referenced papers, and did not find a straight answer there either. The best that I could figure out is from Liang et al. (2019), which stated: “…water-vapor adsorption capacity before deliquescence, such as that of rGTNC microcrystals (7.6 cm3/g) and that of pure NaCl crystal (0.071 cm3/g) at 25% RH”. Please provide here the actual values of absorption factor as a function of RH for the various substances.
Calculating the amount of energy release using an absorption factor of 7.6, and a dry mass of salt of NaCl/TiO2 of 230 g/s, I came up with a calculated added energy of only 17 MW, which is 30% of the thermal heat due to burning the fuel, assuming 60 MW for a fuel burn rate of 1.5 kg/s, which is the consumption of the jet engine. Please provide I the paper the actual value, and don’t let the readers to guestimate it.
The added salt solution is already deliquesced and cannot make much of a difference. If it does absorb more vapor, please show how much vapor is absorbed and how much energy is released per 1-g of each type of salt solution.
Based on that, I conclude that the figure of a factor of 295 does not occur at the deliquescence point, and this explains the lack of realization of the large amounts of hoped latent heating.
The authors assumed a total added energy of condensation of 153 MW, while it is at most 17 MW. Obviously, such a small added heat cannot make much of a difference, as realized in the experimental results.
Figure 1:
Please explain better that the amount of energy is composed of the fuel energy (~60 MW according to my calculations) and the latent heating energy (Pc = various values).
Where is the lifting condensation level? The vertical RH profile is critically important for the rate of condensation on the injected aerosols.
Please plot also the amount of condensed water in g/m3, as this is an essential quantity to the hypothesis.
Lines 277-278: An increase in the humidity will increase the jet lifting only if somehow increased in the jet and not in the ambient air.
Figure 3: Please provide a scale.
Lines 527-528:
How was the height calculated?
There is no height scale on the figures. Please add it.
Lines 529-530:
I could not find any indication for updraft in Fig. 8.
Lines 531-535:
The CCN concentrations should be 3 orders of magnitude larger. Aerosols > 0.3 um are just at the tail end of the distribution. Aerosol concentration near the source of the aerosol dispersion are meaningless.
Figure 7: What are the wavelengths of the images in Figure 7?
Figure 8: Please add a height scale to Figure 8f.
Figure 12: Low layer clouds occur only near the horizon at a location that renders them irrelevant to the experiment.
Please show the position of the plume.
Figure 13: There is no cloud convergence.
The clouds were at different levels moving with different winds.
It seems like the cloud in the last image marked with the lower right arrow was formed in place. What is its position with respect to the plume?
Figure 14: The right picture provides the best insight, showing a diffused cloud composed of deliquescent salt particles.
Figure 15: There is no change in the cloud features that can be ascribed confidently to the seeding.
These rain clouds pre-existed to the seeding.
Figure 16: Please mark the times of the pictures.
The rising cloud base cannot be ascribed to the seeding because it occurred already upwind of the site.
Lack of rain at the site at this time proves that the cloud did not rain-out.
Figure 17:
Please show a sequence of radar images and see if the echo moves with the wind and is not a ground clutter.
Lines 635-636: Thickening was not demonstrated qualitatively or quantitatively.
Line 640: No convergence is demonstrated.
Check if a small cumulus formed at the top of the plume.
Line 641: This scattering cannot be related to the seeding.
Lines 668-669: This is possible if you have enough energy, of course.
Solar energy over 1 km2 is an order of magnitude larger than the experiment energy.
Lines 687-689: This statement is not supported by objective observations.
Lines 690-691: This statement is not supported by objective observations.
Lines 692-695: It is not just a matter of time. The large growth factor at low RH are incorrect. The salt absorption of water can add only a small fraction of energy to the heat from the burning fuel.
Line 699: The statement: “Overall, the obtained effects testify to the correctness of the scientific idea of the method”, is not supported by the observations.
Lines 704-706: This energy can be used much more effectively for water desalination at much larger quantities compared to the most optimal rain enhancement scenario by this method.
Lines 707-709: This principle can provide very little added energy. In fact, fuel can add more energy than any added hygroscopic substance of the same mass. This undermine the usefulness of the whole concept.
Lines 710-712: The rate is not the main issue.
Line 727: This amount of energy can be used for desalinating much more sea water than any single cumulonimbus cloud can produce.
Author Response
Q1. Line 69: The statement is incorrect. For example, numerical weather predictions of the NMC can definitely predict isolated air mass convection over the mountains.
A1. The sentence was modified for “Often, such clouds generate rainfall, grow to the scale of thunderstorms and even hailstorms, although their prediction is still challenging.”
Q2. Lines 154-156: Please provide references for the trials to increase rainfall by creating heat islands.
A2. This information was taken from internet sources. Unfortunately, the publication could not be found. We may remove this offer at your discretion.
Q3. Lines 196-198: When describing the methodology, please define what are the h points. Then quantify the amount of condensed water and released latent heat at the deliquescence point for 1 g of the different types of hygroscopic materials. This is a missing fundamental quantity.
A3. The definition of hygroscopic point was added to the text: “The hygroscopic point (hgp) of hygroscopic solid material can be identified through its water uptake capacity and represents the threshold value of the relative humidity in the air above which the solid substance starts adsorbing water vapor [Tereshchenko 2015].”
A.G. Tereshchenko, “Deliquescence: hygroscopicity of water-soluble crystalline solids,” Journal of Pharmaceutical Sciences, vol. 104, no. 11, pp. 3639–3652, 2015.
Q4. Line 262-263: The value of 295 g of absorbed water per 1 g of NaCl/TiO2 is fantastic. Is it realized already at the deliquescence point or only at the point of RH=100%? I had to go to the referenced papers, and did not find a straight answer there either. The best that I could figure out is from Liang et al. (2019), which stated: “…water-vapor adsorption capacity before deliquescence, such as that of rGTNC microcrystals (7.6 cm3/g) and that of pure NaCl crystal (0.071 cm3/g) at 25% RH”. Please provide here the actual values of absorption factor as a function of RH for the various substances.
A4. This data is taken from the abstract of the article "Yanlong Tai, Haoran Liang, Nabil El Hadri, Ali M. Abshaev, Buzgigit M. Huchinaev, Steve Griffiths, Mustapha Jouiad, and Linda Zou. 2017. Core/Shell Microstructure Induced Synergistic Effect for Efficient Water-Droplet Formation and Cloud-Seeding Application. ACS Nano, https://pubs.acs.org/doi/10.1021/acsnano.7b06114".
Here is a part from this abstract: “We designed and synthesized a novel type of core/shell NaCl/TiO2 (CSNT) particles with controlled particle size, which successfully adsorbed more water vapor (~ 295 times at low relative humidity, 20 % RH) than that of pure NaCl, deliquesced at lower environmental RH of 62 - 66 % than the hygroscopic point (hg.p., 75 % RH) of NaCl, and formed larger water droplets ~ 6 - 10 times of its original measured size area, whereas the pure NaCl still remained as crystal at the same condition. The enhanced performance was attributed to the synergistic effect of the hydrophilic TiO2 shell and hygroscopic NaCl core microstructure, which attracted large amount of water vapor and turned it into liquid faster”.
Q5. Calculating the amount of energy release using an absorption factor of 7.6, and a dry mass of salt of NaCl/TiO2 of 230 g/s, I came up with a calculated added energy of only 17 MW, which is 30% of the thermal heat due to burning the fuel, assuming 60 MW for a fuel burn rate of 1.5 kg/s, which is the consumption of the jet engine. Please provide I the paper the actual value, and don’t let the readers to guestimate it.
A5. The text of the article (new Line 269-275) includes an estimate of the amount of energy released so that the reader does not have to estimate.
The main benefit of using hygroscopic aerosol in a jet is a release of latent heat of vapor condensation higher in the atmosphere (from several hundred meters and higher), where the jet power is no longer so large and energy replenishment is required to maintain buoyancy.
Q6. The added salt solution is already deliquesced and cannot make much of a difference. If it does absorb more vapor, please show how much vapor is absorbed and how much energy is released per 1-g of each type of salt solution.
A6. The idea in spraying aqueous salt solutions of hygroscopic substances was that in this way it is most simple to form an aerosol of a given size if there are nozzles and a pump with the desired characteristics. When leaving the fog cannon, salty droplets up to 15 microns in size (for our cannons) with a known proportion of dissolved salt in them quickly evaporate in the environment of heated air and fuel combustion products and form solid hygroscopic particles. This is confirmed in our test measurements of the aerosol spectrum at a distance of 20-50 m from the cannon output when it was operating horizontally. These and other results are not presented in the article. In this case, the heat lost near the jet engine is not so important. The main benefit should be the release of latent heat of vapor condensation on the formed particles when they rise to several hundred meters and higher, where the jet power is no longer so large and energy replenishment is required to maintain buoyancy.
However, in the second field company, we experimented more with ready-made powders of hygroscopic substances (CaCl2, NaCl, Carbamide) and novel powder.
Q7. Based on that, I conclude that the figure of a factor of 295 does not occur at the deliquescence point, and this explains the lack of realization of the large amounts of hoped latent heating.
A7. We believe that we did not obtain the expected amount of condensation heat (and a level of 295 times) due to insufficient air humidity, which in most cases was below the hygroscopic point of the nanopowder NaCl/TiO2 (NaCl and carbamide too), as well as insufficient speed of condensation processes.
Q8. The authors assumed a total added energy of condensation of 153 MW, while it is at most 17 MW. Obviously, such a small added heat cannot make much of a difference, as realized in the experimental results.
A8. The total added energy of condensation when a nanopowder is introduced into the jet of 260 g/s can be 230, where q = 2.26 kJ/g is the amount of heat that released during the condensation of 1 g of water vapor.
Q9. Figure 1: Please explain better that the amount of energy is composed of the fuel energy (~60 MW according to my calculations) and the latent heating energy (Pc = various values).
Where is the lifting condensation level? The vertical RH profile is critically important for the rate of condensation on the injected aerosols. Please plot also the amount of condensed water in g/m3, as this is an essential quantity to the hypothesis.
A9. The authors agree with the remark. Figure 1 shows the results of calculations of the jet motion with fuel energy (~60 MW) to which the heat of condensation of water vapor on hygroscopic aerosol is added. In this case, it was assumed that the heat of condensation is released continuously, starting from the moment the aerosol is introduced into the rising jet, since the aerosol was introduced continuously. In order to simplify the calculations, the air humidity at all heights was assumed to be 80%, i.e. above the hygroscopic points of all types of aerosols introduced into the jet. The dependence of the amount of condensate on the vertical moisture profile has not been studied.
The purpose of solving such a simplified problem was to test the performance of the physical principle of the method being tested, which differs from the previously tested meteotrons of Dessence, Vulfson and Levin, etc. in that the energy of the jet is supplemented by the heat of condensation. The main attention is paid to assessing whether such jet energetic replenishment can increase its ascent height and the potential of the meteotron in creating artificial clouds, as well as how the amount of replenishment affects at different vertical gradients of temperature and wind speed.
Q10. Lines 277-278: An increase in the humidity will increase the jet lifting only if somehow increased in the jet and not in the ambient air.
A10. The remark is fair. This sentence has been revised as: “An increase in air humidity also leads to an increase in the jet lifting height due to condensation of water vapor”.
In addition, 1.5 liter/s of water is injected at the inlet of the air intake of a jet engine to increase compression and engine power. The spraying of aqueous solutions of hygroscopic substances in a total amount of about 3.3 l/s also increases the air humidity in the jet. In total, about 4.0 liter/s is injected into the jet. This means that the actual moisture content of the jet is increased by approx. 20 g/m3. But in the process of ascent, a high-speed jet entrains significant volume of ambient air, which leads to an equalization of humidity in the jet and the surrounding air.
Q11. Figure 3: Please provide a scale.
A11. Added dimensions for the length, height and diameter of the outlet nozzle.
Q12. Lines 527-528: How was the height calculated? There is no height scale on the figures. Please add it.
A12. The height was determined approximately by the location of the upper point of the jet plume and the top of the nearby mountain, the height of which was around 250-300 m (see Fig. 6). Next, we approximated the visual plume of the jet obtained by a video camera and a thermograph in proportion to the distance from the top of the mountain. Unfortunately, we didn't have a better reference point.
The height of the jet rise, according to the IRTIS-2000-C thermograph, reached approximately 600-700 m with a maximum of about 1100-1200 m above the surface (see Fig. 527 7 and Fig. 8f).
Q13. Lines 529-530: I could not find any indication for updraft in Fig. 8.
A13. The simulated shape of the updraft is shown in fig. 8e. The shape of the updraft, measured by a thermograph in the range of 3-5 microns, is shown in fig. 8f. The flow detected visually with a video camera is shown in 8g-8h. The instrumental remote and in-situ measurement of the flow velocity is associated with a number of technical difficulties; in this case, we were unable to make detailed measurement using the means available to us.
Q14. Lines 531-535: The CCN concentrations should be 3 orders of magnitude larger. Aerosols > 0.3 um are just at the tail end of the distribution. Aerosol concentration near the source of the aerosol dispersion are meaningless.
A14. It is a fair remark. Instead of concentration in cubic meters, the y-axis should be cubic liter. The labels on the vertical axes have been corrected in Figures 9 and 10 as well as in the text in Lines 531-535.
In the second part, we are not focusing on cloud condensation nuclei concentration, but on the background aerosol in the environment, measured using a Particle Counter DT-9880M (www.cem-instruments.com).
It should also be taken into account that the experimental site was at an altitude of 1600 m above sea level, which is located at the upper boundary of the boundary layer, where the content of air impurities should be lower than at sea level.
Figure 10 has been removed from the article.
Q15. Figure 7: What are the wavelengths of the images in Figure 7?
A15. Spectral range of IRTIS-2000 thermograph is 3-5 µm. This information has been added to the text in Table 1.
Q16. Figure 8: Please add a height scale to Figure 8f.
A16. The height scale in fig. 8e and 8f are the same. In figure 8e, the lines of the grid of heights have been drawn.
Q17. Figure 12: Low layer clouds occur only near the horizon at a location that renders them irrelevant to the experiment. Please show the position of the plume.
A17. We agree that the evolution of this cloud may not be associated with the action of a jet and a high concentration of hygroscopic aerosol. Therefore, the text about effect 1 and figure 11 have been removed in the new edition of the article.
Q18. Figure 13: There is no cloud convergence.
The clouds were at different levels moving with different winds.
It seems like the cloud in the last image marked with the lower right arrow was formed in place. What is its position with respect to the plume?
A18. In the figure 13, it was not possible to fully convey the effect of convergence, since this is a timely extended dynamic process. Such an effect was indeed observed from different angles, both by means of photo-video recording and by observers. This figure shows that the clouds located on the left and right are moving towards each other. The plume was directed into the cloudless sky between the clouds. After the aircraft engine stopped, the convergence of small cloud formations was not observed; they drifted approximately equally in the wind. However, we cannot confidently state that this was the result of our influence. This may also be due to the convergence of air streams from the Arabian and Oman Gulfs, interacting with each other and with the specific orography at the place of the experiments.
Q19. Figure 15: There is no change in the cloud features that can be ascribed confidently to the seeding. These rain clouds pre-existed to the seeding.
A19. Figure 15 clearly shows the darker plumes of engine jet with aerosols and light zones above where jet entered the cloud. In all experiments, when clouds passed at a low altitude or through the position of field experiments, we observed the opening of clouds (visibility increase or clearing) in the propagation zone of a jet saturated with hygroscopic aerosol. Here, several factors should be expected to act simultaneously: 1) dynamic, when a gas-aerosol jet blows through the cloud volume, pushing cloud drops out of the jet; 2) turbulent, when the turbulences that occur during the propagation of a gas-aerosol jet enhance the coagulation growth of cloud drops; 3) thermal, when the heated gas emitted by an aircraft engine evaporates cloud drops; 4) condensation-coagulation, when an artificial aerosol, including a coarse one, causes microphysical processes environment (condensation, coagulation) between cloud drops and small droplets and hygroscopic aerosol. This allows us to hope that this effect is not caused by chance, but as a result of the experimental setup. We agree that the entire chain of these processes can be studied in more detail, for example, with the help of acceptable numerical models, if any.
Q20. Figure 16: Please mark the times of the pictures.
The rising cloud base cannot be ascribed to the seeding because it occurred already upwind of the site.
Lack of rain at the site at this time proves that the cloud did not rain-out.
A20. The time is indicated directly on the photographs of the drawing, local time is 15-00, 15-01, 15-02 and 15-04, 12/26/2021.
Dispersion (evaporation) of the cloud began when it flowed onto the jet near the earth's surface, and also from 15:00 to 15:04, and from 15:05 at a certain height on the windward flank, where the jet was deflected by the surface wind. Complete dissipation of the cloud occurred in less than 4 minutes. We believe that this happened under the influence of the jet.
Q21. Figure 17: Please show a sequence of radar images and see if the echo moves with the wind and is not a ground clutter.
A21. We obtained radar information from www.ncm.ae. At the locations shown in Figures 17, no radio echo was observed before and after (neither from interference from the ground nor from clouds). Therefore, we assume that the appearance of these radio echoes may be associated with the action of the jet and hygroscopic reagents.
Q22. Lines 635-636: Thickening was not demonstrated qualitatively or quantitatively.
A22. We agree with this remark. Aerial instrumental measurements and observations from above (from an airplane, helicopter or drone) are required for a confident statement. However, at the experimental position, located near the border of the UAE and Oman, flights and aerial photography were prohibited by the military bodies.
But in this place, it meant compaction with subsequent enlightenment in the volume of the cloud where the jet propagated. Sentence was changed to: “3 cases – compaction and further formation of the enlightenment zone of clouds into which a jet saturated with hygroscopic aerosols was injected”.
Q23. Line 640: No convergence is demonstrated. Check if a small cumulus formed at the top of the plume.
A23.The answer to this question is similar to answer A18
Q24. Line 641: This scattering cannot be related to the seeding.
A24. We do not agree with this statement, but respect the opinion of the reviewer. There are a number of papers in the literature on the dynamic impact on various drop-aerosol media, for example, the impact of a thermal jet on fogs in order to disperse it. For example:
Page 14, 48
https://patents.google.com/patent/US3227373
You can find there statements about dispersion of fog conditions and also making the cloud medium precipitable through engine jet forcing.
Q25. Lines 668-669: This is possible if you have enough energy, of course.
Solar energy over 1 km2 is an order of magnitude larger than the experiment energy.
A25. We agree with this statement. The main issue is the resources needed to multiply the updraft area. But even solar meteortrons have weaknesses, since convective jets are quite strongly displaced by the wind even at low altitude. In the case of mountain-valley circulation, at least 2 effects work - dynamic due to the forced rise of air and thermal when the surface of the peaks is heated (due to sharp gradient of air temperature over mountain surface and ambient atmosphere at the same height).
Q26. Lines 687-689: This statement is not supported by objective observations.
A26. Our visual observations just showed the formation of clearing zones, which, in turn, could not but change the microphysical state of the cloud in the volume of jet propagation. We partially answered this in A19.
Q27. Lines 690-691: This statement is not supported by objective observations.
A27. This statement has been answered above.
Q28. Lines 692-695: It is not just a matter of time. The large growth factor at low RH are incorrect. The salt absorption of water can add only a small fraction of energy to the heat from the burning fuel.
A28. We do not fully agree with the comment. We reformulated this phrase as follows: “The reason for the insufficient effectiveness of the method is that the low air humidity excluded the condensation of water vapor on the types of hygroscopic aerosols used, and also that the condensation process is rather slow.”
At the same time, it should be borne in mind that the idea of ​​adding a hygroscopic aerosol to the jet is to increase its buoyancy due to the heat of water vapor condensation on artificial aerosols at altitudes where the jet energy of the aircraft engine is already low. At such altitudes, the energy of condensation heat can be not only comparable, but also significantly exceed the energy of a pure jet.
Q29. Line 699: The statement: “Overall, the obtained effects testify to the correctness of the scientific idea of the method”, is not supported by the observations.
A29. We see no contradiction in this phrase with what has been received. Both theory and practice have shown their adequate agreement. Approximate jet heights in windy and calm situations, the negative effect of the wind breaking blowing the jet to the side were predicted by the model and confirmed in practice. By analogy negative impact of low humidity and weak temperature lapse rates accompanied with blocking layers were all predicted by the model as well. Under certain meteorological situations, a plume of droplets formed on particles was observed. But it turned out that the scale of our action is insufficient and future works can be devoted to the system scaling up issue.
Q30. Lines 704-706: This energy can be used much more effectively for water desalination at much larger quantities compared to the most optimal rain enhancement scenario by this method.
A30. The energy and financial resources spent on the seawater desalination program around the world are disproportionately greater. At the same time, our previous published studies, mentioned above, have shown that the water content of one powerful cumulonimbus cloud can exceed 10^7 tons. Such a cloud can produce a layer of precipitation 30-50 mm thick over an area of 10x30 km, which is about 10^7 m-3 of rainwater. This amount of water can be produced by the most powerful seawater desalination plants in the United Arab Emirates and Saudi Arabia in 10 days. At the same time, the cost of such a desalination plant is 2 billion US dollars, excluding the cost of electricity and other operating expenses. Secondly, it seems to us expedient to continue the search for alternative technologies.
Q31. Lines 707-709: This principle can provide very little added energy. In fact, fuel can add more energy than any added hygroscopic substance of the same mass. This undermine the usefulness of the whole concept.
A31. The answer is given in the previous paragraphs. The heat energy of condensation can exceed the energy of the jet when it decreases many times as it rises in the atmosphere.
Q32. Lines 710-712: The rate is not the main issue.
A32. We changed this sentence in the text to the following: “Improving the speed of the used hygroscopic agents (as well as glaciogenic agents) is no less important for the practice of weather modification than optimizing the temperature and humidity of the air at which they work. The speed and temperature-humidity regime of seeding materials are one of the most important parameters affecting the efficiency of cloud seeding».
Q33. Line 727: This amount of energy can be used for desalinating much more sea water than any single cumulonimbus cloud can produce.
A33. According to open data the specific energy consumption of thermal desalination plants varies widely – from 7 to 27 kWh/m3, depending on the technology. Our previous analysis (see ref. below) showed that typical hailstorm cloud in Northern Caucasus produces 10^4 – 5×10^5 ton of rain water per minute. This amount of water is comparable to the amount of water flowing in such large rivers as, for example, the Volga, Danube, Dnieper, etc.
That volume of potable water to be produced using desalination needs about 70 – 13500 MW of energy per minute or 4,2 – 810 GWh, while total energy capacity of all desal. plants by 2017 was only around 12 GWh. That is much beyond the power of all desalination plants. Therefore single storm produces 3 – 30 times more rainwater per time unit then all currently constructed desalination plants.
Abshaev, M.T., Abshaev, A., Malkarova, A.M& Mizieva, Z. (2009). Radar estimation of water content in cumulonimbus clouds. Journal of Izvestiya – Atmospheric and Ocean Physics 45, 731–-736. https://doi.org/10.1134/S0001433809060061.
Abshaev, A.M., Flossmann, A., Siems, S.T., Prabhakaran, T., Yao, Z., Tessendorf, S. (2022). Rain Enhancement Through Cloud Seeding . In: Qadir, M., Smakhtin, V., Koo-Oshima, S., Guenther, E. (eds) Unconventional Water Resources . Springer, Cham. https://doi.org/10.1007/978-3-030-90146-2_2
Authors express their gratitude to the reviewer for fair remarks. Their correction led to the improvement of the manuscript.
Round 2
Reviewer 2 Report
The authors seem to have put a lot of effort into my comments. However, there are still many parts that need to be fixed.
The manuscript is still not well structured. Paragraphs should not be simply listed but described in organic connection with each other. This cannot be solved by simply adding some content, and many parts of the manuscript must be rewritten. For example, the introduction ends with a description of the shortcomings of the preceding methods. After this, it should be stated whether this shortcoming can be overcome in this study. The introductory paragraph for each session added at the end to the introduction is very good. However, that paragraph should describe all sessions.
The size of the figures and tables still seems to exceed the size of the manuscript.
According to the author's answer, hygroscopic point seems to mean deliquescence relative humidity (DRH). Once the relative humidity exceeds the DRH, an aqueous solution is formed, efflorescence (recrystallization) of the liquid do not occur until efflorescence relative humidity (ERH). In this study, when the aerosol meets the jet, doesn't it encounter higher humidity than DRH? I wonder if hygroscopic point is important in this study.
Author Response
Q1. The manuscript is still not well structured. Paragraphs should not be simply listed but described in organic connection with each other. This cannot be solved by simply adding some content, and many parts of the manuscript must be rewritten. For example, the introduction ends with a description of the shortcomings of the preceding methods. After this, it should be stated whether this shortcoming can be overcome in this study. The introductory paragraph for each session added at the end to the introduction is very good. However, that paragraph should describe all sessions.
A1. Reviewer's comment is accepted. The text was modified accordingly. The content of each section and their organic interconnections are improved. The new text is highlighted in blue.
Q2. The size of the figures and tables still seems to exceed the size of the manuscript.
A2. We agree with the comment. In the new version of the article, the large figures and tables have been redesigned and their sizes have been reduced.
Q3. According to the author's answer, hygroscopic point seems to mean deliquescence relative humidity (DRH). Once the relative humidity exceeds the DRH, an aqueous solution is formed, efflorescence (recrystallization) of the liquid do not occur until efflorescence relative humidity (ERH). In this study, when the aerosol meets the jet, doesn't it encounter higher humidity than DRH? I wonder if hygroscopic point is important in this study.
A3. Indeed, we consider the hygroscopic point (hgp) to be the value of relative humidity (DRH) above which water vapor begins to condense and form an aqueous solution. When meeting with a jet of heated gases having a temperature of about 200-300 degrees Celsius, Drops of aqueous solutions evaporate(drops containing dissolved hygroscopic salts inside). Further, as the jet rises, the temperature drops, and at some point in time, supersaturation of water vapor density over surface of hygroscopic aerosols may occur. Here the hygroscopic point is very important, determining at what height from the experimental setup condensation will begin supplying latent heat of condensation, which feeds jet buoyancy. Thus, it is assumed that water droplets will form below the cloud base level, or the level at which droplet formation begins on the natural aerosol present in the area. This can only be achieved by using highly effective hygroscopic materials that absorb water vapor at low water vapor densities (at RH < 100%). In this regard, the hygroscopicity of the aerosol is of decisive importance. But, unfortunately, it turned out that in such an arid region as the UAE, air humidity is usually very low.
Reviewer 3 Report
The hole basis of the study relies on the authors assertion that the water mass absorption factor is 295. However, a careful reading of the reference n which the authors rely reveals that the water mass absorption factor is only 7.5. This is only 2.5% of the claimed effect.
More specifically, Figure 2 of Hadri et al. (2017) shows the real numbers. The factor of 295 is for the ratio of water uptake of CSNT/Nacl at 25%, but it is for very small absolute amount of vapor of up to 50 cm-3 g-1, which is ~4% by weight. The vapor absorption at relative humidity of 75% increases the surface area of CSNT by a factor of 8, which means a volume factor of 22.6. For specific density of CSNT of ~3 g/cm-3, this translate to an mass absorption factor of ~7.5 g of water per 1 g of CSNT, and not 295 g as used by the authors. It appears that the whole premise of this research line is based on misreading of Hadri et al. (2017).
Clarifying this misunderstanding is a prerequisite for publishing this paper.
There are many additional issues with non acceptance of my previous comments. The authors must accept the reality in my first comment, and address respectively my other comments before the paper can be considered for publication.
I strongly urge the authors to publish the paper with the correct effects and the nil results, because the community has been misled by the misconception of the large absorption condensation effect.
Author Response
Q1. The hole basis of the study relies on the authors assertion that the water mass absorption factor is 295. However, a careful reading of the reference in which the authors rely reveals that the water mass absorption factor is only 7.5. This is only 2.5% of the claimed effect.
More specifically, Figure 2 of Hadri et al. (2017) shows the real numbers. The factor of 295 is for the ratio of water uptake of CSNT/Nacl at 25%, but it is for very small absolute amount of vapor of up to 50 cm-3 g-1, which is ~4% by weight. The vapor absorption at relative humidity of 75% increases the surface area of CSNT by a factor of 8, which means a volume factor of 22.6. For specific density of CSNT of ~3 g/cm-3, this translate to an mass absorption factor of ~7.5 g of water per 1 g of CSNT, and not 295 g, as used by the authors. It appears that the whole premise of this research line is based on misreading of Hadri et al. (2017).
Clarifying this misunderstanding is a prerequisite for publishing this paper.
A1. We accept this remark. All estimates related to the amount of condensed moisture have been removed from the text of the article, including fig. 1, where the calculated data are presented for some assumed values of the jet energy replenishment. All other parts of the article are not related to the specified coefficient (295).
Q2. There are many additional issues with non acceptance of my previous comments. The authors must accept the reality in my first comment, and address respectively my other comments before the paper can be considered for publication.
A2. The authors of the article paid great attention to all 32 comments of the distinguished reviewer. Most of the remarks have been eliminated, some comments have been given. Appropriate changes have been made to the text of the article, the design of figures and tables.
Q3. I strongly urge the authors to publish the paper with the correct effects and the nil results, because the community has been misled by the misconception of the large absorption condensation effect.
A3. Agree that the effects of water vapor condensation on NaCL/TiO2 nanopowder were overestimated. In the new version of the paper this has been eliminated. Moreover, the field experiments show that the effect of condensation heat on the jet characteristics is not very significant, and is absent in low humidity conditions at all altitudes.
Round 3
Reviewer 2 Report
It seems that the authors have taken my comments well. I think that this manuscript can be accepted.
Author Response
We express our deep respect to the reviewer for all insightfull remarks and recomendations. Their correction led to the improvement of the manuscript.
Reviewer 3 Report
The authors accept that the water mass absorption factor of the NaCL/TiO2 is only 7.5. But the paper does not reflect fully this acceptance.
The study still relies on a recent reference 35 (Abshaev et al., Scientific Reports 2022), where the authors again use the salt mass expansion ratio of near 295, erroneously. All references to that erroneous study must be deleted and the conclusions should be changed respectively.
Here is a quote from that paper (Abshaev et al., Scientific Reports 2022) proving that it is based on that error:
“The introduction of NaCl/TiO2 micro powder with k1 = 295 into the jet at 230 g/s26 can lead to the condensation of water vapor in the amount M′
wt = N・ma ・ k1 = 2 ・ 1011 ・ 1.133 ・ 10−9 ・ 295 ≈ 66.8kg/s.”
The authors should have used the correct values. They wrote in lines 239-241:
“The results of estimating the amount of condensation heat that can be released during the massive introduction of three types of aerosols into the jet stream (Table 1), and the results of 3-D modeling of the jet motion under different atmospheric conditions showed very encouraging results [35].”
But Table 1 does not list these values, which are anyway erroneous as they still rely on the wrong calculations in reference [35].
The authors must state clearly the correct amount of added condensation and the fraction that it adds to the generated heat due to burning fuel in each of the experiments, as this is the core of the hypothesis of this seeding experiment.
Due to the importance, the range of this percentage has to be mentioned also in the abstract.
This assertion has no basis in the findings., and has to be deleted.
The conclusions of the paper do not support the stated optimism with respect to this method. The calculations that provide more optimism are erroneous, as stated above.
This is evident in lines 24-26 in the abstract state: “Nevertheless, a set of optimization metrics were derived based on the experimental results and preliminary simulations suggest the technology, with the recommended adjustments, may be effective for creating artificial clouds and precipitation.”
Author Response
Q1. The authors accept that the water mass absorption factor of the NaCL/TiO2 is only 7.5. But the paper does not reflect fully this acceptance.
A1. We do not agree and do not admit that “the water mass absorption factor of the NaCL/TiO2 is only 7.5” and do not see the need to make such a recognition in the article. Even for pure NaCL, this coefficient is almost 2 times greater than 7.5. To prove this, consider the results of two laboratory experiments described in [1 and 2]:
- According to laboratory experiments (Ludlam, 1950), NaCl particles weighing 10-9 g at 100% air humidity form drops 30 microns in diameter within a few seconds.
In this experiment, the initial characteristics of the NaCl particle are:
Weight ma = 10-9 g, density ρa = 2.165 g/cm3, da= 9,59 micron
Droplet formed on NaCl particle has parameters:
Diameter dt = 30 micron = 3×10-3 cm, volume Vt = π*dt3 / 6 = ;
Weight mt = ma + mw = ma + π*dt3*ρw/6 = 10-9 + 3.14*(3*10-3)3*1,0/6 = 15.13*10-9g;
Density ρt = mt/Vt = 15.13*10-9 g / 14.13*10-9 cm3 = 1.07 g/cm3
Weight of condensed water (on NaCl) mw = mt – ma = 14.13ma.
Therefore, according to experiment of famous F.H. Ludlam for NaCl aerosol the coefficient of water vapor absorption is k1 = mw/ma = 14.13.
- In the figure attached below from the article [2], one can see that in the cloud chamber at a temperature of 5 °C and a humidity of 100%, drops formed on a NaCl/TiO2 particle were about 2.75 times in diameter, and 20.8 times in volume larger, than on a NaCl particle of the same size.
This means that a NaCl/TiO2 particle can condense 20.8 times more water vapor than a NaCl particle.
Taking into account the data of both experiments, it follows that the NaCl/TiO2 particle can condense water vapor in the amount n3 = n1 × n2 = 20.8×14.13 = 293.9 times its dry weight.
For a more accurate answer, more accurate measurements should be made in the second experiment, but this is not the subject of the article under discussion. In it, we are only interested in the possibility of increasing the buoyancy of the jet by replenishing it with the heat of condensation. Numerical modeling has shown that, in principle, this can increase the jet energy and the height of its rise, but experiments have shown that such replenishment almost does not work due to low air humidity in the UAE and the slowness of the process of water vapor condensation.
References
- Ludlam F.H. The composition of coagulation elements un cumulonimbus. Quart. J. Roy. Meteor. Sos., Vol. 76, No 327, 1950.
- Haoran Liang, M.T. Abshaev, A.M. Abshaev, B.M. Huchunaev, Steven Griffiths, Linda Zou, 2019: Water vapor harvesting nanostructures through bioinspired gradient-driven mechanism. Chemical Physics Letters 2019, 728, 167-173. https://doi.org/10.1016/j.cplett.2019.05.008
Q2. The study still relies on a recent reference 35 (Abshaev et al., Scientific Reports 2022), where the authors again use the salt mass expansion ratio of near 295, erroneously. All references to that erroneous study must be deleted and the conclusions should be changed respectively. Here is a quote from that paper (Abshaev et al., Scientific Reports 2022) proving that it is based on that error: “The introduction of NaCl/TiO2 micro powder with k1 = 295 into the jet at 230 g/s26 can lead to the condensation of water vapor in the amount M′
wt = N・ma ・ k1 = 2 ・ 1011 ・ 1.133 ・ 10−9 ・ 295 ≈ 66.8kg/s.”
The authors should have used the correct values. They wrote in lines 239-241:
“The results of estimating the amount of condensation heat that can be released during the massive introduction of three types of aerosols into the jet stream (Table 3), and the results of 3-D modeling of the jet motion under different atmospheric conditions showed very encouraging results [35].”
A2. Based on the response to remark Q1, the authors cannot agree with the assertion that the coefficient k1 = 295, the error of the study (Abshaev et al., Scientific Reports 2022, [35]), and see no reason to make the corrections recommended by the reviewer. Table 3 has been restored in full.
Q3. But Table 1 does not list these values, which are anyway erroneous as they still rely on the wrong calculations in reference [35].
A3. In accordance with the answer to remark Q1, the authors cannot agree with the incorrectness of the calculations in [35].
Q4. The authors must state clearly the correct amount of added condensation and the fraction that it adds to the generated heat due to burning fuel in each of the experiments, as this is the core of the hypothesis of this seeding experiment.
A4. The authors believe that the amount of condensate and the proportion of heat generated by the experimental setup are correct.
Q5. Due to the importance, the range of this percentage has to be mentioned also in the abstract. This assertion has no basis in the findings., and has to be deleted.
A5. The authors agree with this. Some corrections have been introduced in the annotation, which are highlighted in blue.
Q6. The conclusions of the paper do not support the stated optimism with respect to this method. The calculations that provide more optimism are erroneous, as stated above.
A6. The authors agree that the experiments did not confirm the expected increase in jet buoyancy due to the introduction of a hygroscopic aerosol into the jet. Appropriate changes have been made at the end of the article.
Q7. This is evident in lines 24-26 in the abstract state: “Nevertheless, a set of optimization metrics were derived based on the experimental results and preliminary simulations suggest the technology, with the recommended adjustments, may be effective for creating artificial clouds and precipitation.”
A7. In view of this remark, the specified phrase has been removed from the text

Round 4
Reviewer 3 Report
The authors do not accept the fact that their methodology is based on an error.
To support their claim, they show that NaCl expansion is very large, but the study quoted by the author shows it at RH=100%. But this would mean nothing as at 100% cloud water condenses anyway. The premise of the method is that NaCL/TiO2 expands x295 at much lower RH, which is demonstrated to be not true.
In a previous review I already wrote:
Figure 2 of Hadri et al. (2017) shows the real numbers. The factor of 295 is for the ratio of water uptake of CSNT/Nacl at 25%, but it is for very small absolute amount of vapor of up to 50 cm-3 g-1, which is ~4% by weight. The vapor absorption at relative humidity of 75% increases the surface area of CSNT by a factor of 8, which means a volume factor of 22.6. For specific density of CSNT of ~3 g/cm-3, this translate to an mass absorption factor of ~7.5 g of water per 1 g of CSNT, and not 295 g, as used by the authors. It appears that the whole premise of this research line is based on misreading of Hadri et al. (2017).